# META-ROUTER: BRIDGING GOLD-STANDARD AND PREFERENCE-BASED EVALUATIONS IN LLM ROUTING

**Yichi Zhang**[1,*]    **Fangzheng Xie**[1]    **Shu Yang**[2]    **Chong Wu**[3,4,*]

[1] Department of Statistics, Indiana University Bloomington
[2] Department of Statistics, North Carolina State University
[3] Department of Biostatistics, The University of Texas MD Anderson Cancer Center
[4] Institute for Data Science in Oncology, The University of Texas MD Anderson Cancer Center

## ABSTRACT

In language tasks that require extensive human–model interaction, deploying a single "best" model for every query can be expensive. To reduce inference cost while preserving the quality of the responses, a large language model (LLM) router selects the most appropriate model from a pool of candidates for each query. A central challenge to training a high-quality router is the scarcity of reliable supervision. Gold-standard data (e.g., expert-verified labels or rubric-based scores) provide accurate quality evaluations of LLM responses but are costly and difficult to scale. In contrast, preference-based data, collected via crowdsourcing or LLM-as-a-judge systems, are cheaper and more scalable, yet often biased in reflecting the true quality of responses. We cast the problem of LLM router training with combined gold-standard and preference-based data into a causal inference framework by viewing the response evaluation mechanism as the treatment assignment. This perspective further reveals that the bias in preference-based data corresponds to the well-known causal estimand: the conditional average treatment effect (CATE). Based on this new perspective, we develop an integrative causal router training framework that corrects preference-data bias, address imbalances between two data sources, and improve routing robustness and efficiency. Numerical experiments demonstrate that our approach delivers more accurate routing and improves the trade-off between cost and quality. Illustrative code to reproduce our main experiment is available at https://github.com/yichistat/Meta-router.

## 1 INTRODUCTION

As LLM deployments scale and model size grow, serving every request with the strongest model becomes economically and operationally impractical for a commercial success of AI applications. LLM routing (Ding et al., 2024; Hu et al., 2024; Ong et al., 2024) addresses this issue by constructing a decision framework that assigns each incoming query either to larger, more powerful models or to cheaper but potentially weaker ones, thereby balancing cost and performance trade-offs. Traditional cascading routers sequentially process a query through a series of LLMs, from light to heavy, until a satisfactory response is obtained (Chen et al., 2024), but this approach is often inefficient and introduces latency from repeated calls. Predictive routers (Ong et al., 2024; Stripelis et al., 2024; Somerstep et al., 2025; Tsiourvas et al., 2025) instead predict the appropriate model in one shot, often by learning a mapping from query feature (such as text embeddings) to a target model under a cost-quality objective using statistical and machine learning (ML) methods. Another important line of work uses confidence- or reward-model-based routing (Chuang et al., 2025; Frick et al., 2025; Wu & Lu, 2025), which selects models based on uncertainty estimates or learned reward scores associated with each candidate response.

The effectiveness of predictive routers critically depends on the evaluation metrics available in the training data. Existing works differ in the evaluation mechanisms used. For example, Ong et al. (2024) use the LMArena dataset (Chiang et al., 2024), where model preference are judged by internet

---

*Correspondence: Yichi Zhang (yiczhan@iu.edu) & Chong Wu (cwu18@mdanderson.org)

users, and further combine it with standardized benchmarks such as MMLU (Hendrycks et al., 2020) or with LLM-judge-labeled datasets. In contrast, Tsiourvas et al. (2025); Stripelis et al. (2024) employ accuracy-based benchmarks where queries admit objectively verifiable solutions.

In this work, we consider the LLM routing problem in challenging yet realistic scenarios, where humans and LLMs have complex interactions within expert knowledge domains, such as professional healthcare conversations, AI-assisted programming, and exploratory scientific research. In these scenarios, the queries are often open-ended, so accurate evaluation often require domain expertise, multi-criterion rubrics, and careful inspection, making gold labels both costly and labor-intensive to acquire (Chang et al., 2024). This partially explains why the sample size of benchmark datasets of different professional domains with carefully designed evaluation is small. For example, the sample size of HealthBench (Arora et al., 2025) designed for healthcare dialogue is 5000. These challenges hinder the efficient training of routers with sufficient and high-quality samples. Although crowdsourcing or LLM-as-a-judge systems may offer scalable alternatives, such evaluations can be systematically biased relative to expert judgments or task-specific rubrics and may not reliably reflect the true quality of responses (Zheng et al., 2023a; Tam et al., 2024).

These limitations highlight the need for a principled method that can integrate scarce but accurate gold-standard data with scalable yet potentially biased preference-based data efficiently, for debiased LLM router training. We address this challenge from a novel angle by casting it into a causal inference framework, where the response evaluation mechanism is viewed as the treatment assignment. This perspective links router training and debiasing to the extensive literature on semiparametric causal estimation (Imbens & Rubin, 2015; Chernozhukov et al., 2018), and further shows that the bias in preference-based data corresponds to the conditional average treatment effect (CATE), which can be efficiently estimated via causal meta-learners (Künzel et al., 2019). Building on this insight, we propose a meta-router training framework that corrects preference-data bias through R- and DR-learners for CATE estimation (Nie & Wager, 2021; Kennedy, 2023), thereby mitigating sample imbalances across heterogeneous data sources and enabling robust, efficient routing decisions, particularly in human–AI interaction scenarios within high-expertise fields.

## 2 LLM ROUTING WITH GOLD-STANDARD AND PREFERENCE-BASED DATA

LLM responding process towards a human query can be mathematically represented as a (random) function $\mathcal{M} : \mathcal{Q} \mapsto \mathcal{A}$ mapping any query $q \in \mathcal{Q}$ to an answer $\mathcal{M}(q) \in \mathcal{A}$. Here, $\mathcal{Q}$ and $\mathcal{A}$ are the text spaces of queries and answers, respectively. For simplicity, in this work, we consider pairwise LLM routing between two models, namely $\mathcal{M}_p$ and $\mathcal{M}_a$, where $\mathcal{M}_p$ denotes a premium model with generally higher response quality (*e.g.*, GPT-5 (OpenAI, 2025)), and $\mathcal{M}_a$ represents its cost-effective alternative with lower inference cost but potentially lower response quality for certain queries (*e.g.*, GPT-4o mini (OpenAI, 2024)). Given a query $q$, the router learns a policy $\pi(q) \in \{M_p, M_a\}$ that maximizes expected utility function involving inference cost and response quality.

### 2.1 GOLD-STANDARD AND PREFERENCE-BASED DATA

We refer to *gold-standard data* (GS data) as the high-quality dataset, where response quality is assessed either by domain experts or by "gold labels" (Hendrycks et al., 2020; Arora et al., 2025). Hence, it is generally considered the authoritative ground truth for LLM response evaluation. We consider the GS data in the form of

$$\mathcal{D}_G = \{(q_i, r_i)\}_{i=1}^n,$$

where $q_i$ denotes the $i$th query and $r_i$ represents the evaluated quality gain between $\mathcal{M}_p(q_i)$ and $\mathcal{M}_a(q_i)$ under the gold standard. Without loss of generality, we assume that $r_i > 0$ indicates $\mathcal{M}_p(q_i)$ outperforms $\mathcal{M}_a(q_i)$, $r_i < 0$ indicates the opposite, and a value near 0 suggests comparable quality. For example, when correctness is objectively defined (*e.g.*, the MMLU dataset), we define $r_i = 1$ if $\mathcal{M}_p(q_i)$ answers correctly and $\mathcal{M}_a(q_i)$ does not, $r_i = 0$ if both are correct or both are incorrect, and $r_i = -1$ when only $\mathcal{M}_a(q_i)$ is correct. As another example, when $r_i$ is evaluated by domain experts, the expert typically rates $\mathcal{M}_p(q_i)$ and $\mathcal{M}_a(q_i)$ respectively, based on predefined scoring rubrics, and $r_i$ is defined as the difference between these ratings.

We consider the standard probabilistic modeling for the generation of $\mathcal{D}_G$. In particular, we assume $(q_1, r_1), \ldots, (q_n, r_n)$ are independent and identically distributed (iid) generated with $q_i \sim \mathcal{Q}$ for

some query distribution $\mathcal{Q}$, and

$$r_i = \psi(q_i) + \epsilon_i, \tag{1}$$

where the random errors $(\epsilon_i)_{i=1}^n$ satisfy $\mathbb{E}(\epsilon_i \mid q_i) = 0$, and $m : \mathcal{Q} \mapsto \mathbb{R}$ is the average quality gain of some GS model.

Despite their high accuracy, GS data are typically labor-intensive to obtain and difficult to scale. For open-ended queries, response evaluation often requires expert judgment or carefully designed scoring rubrics, particularly in domain-specific professional contexts. Conversely, if only queries with clear standard answers (*e.g.*, the MMLU dataset) are retained, the empirical distribution of $(q_i)_{i=1}^n$ may fail to adequately represent the queries encountered in daily practice.

On the other hand, the *preference-based evaluation* offers a more scalable yet typically more subjective alternative for assessing LLM responses. For instance, LMArena (Chiang et al., 2024) evaluates the LLM responses based on Internet users' preferences, while the LLM-as-a-judge system employs an LLM to directly compare and grade LLM responses (see, *e.g.*, §3.1 in Zheng et al. (2023a)).

Specifically, we denote the *preference-based data* (PB data) by $\mathcal{D}_P = \{(q_i', y_i)\}_{i=1}^m$, where $q_i' \sim \mathcal{Q}'$ denotes the $i$th query from distribution $\mathcal{Q}'$, and $y_i$ represents the outcome of comparing the responses from $\mathcal{M}_p(q_i')$ and $\mathcal{M}_a(q_i')$ through a preference-based mechanism. Similar to $\mathcal{D}_G$, we assume the samples in $\mathcal{D}_P$ are iid and

$$y_i = \eta(q_i') + \epsilon_i', \tag{2}$$

where the random errors $(\epsilon_i')_{i=1}^m$ satisfy $\mathbb{E}(\epsilon_i' \mid q_i') = 0$, and $\eta : \mathcal{Q} \mapsto \mathbb{R}$ is the average quality gain under a preference-based evaluation mechanism. Preference-based evaluation mechanisms are usually simple and intuitive. For instance, the pairwise comparison in an LLM-as-a-judge system or LMArena, returns $y_i = 1$ if $\mathcal{M}_p(q_i')$ is preferred over $\mathcal{M}_a(q_i')$, $y_i = -1$ if the opposite holds, and $y_i = 0$ in the case of a tie. There are multiple approaches to model the preference data generation and $\eta(q)$, *e.g.*, the Bradley-Terry-Luce (BTL) model (Bradley & Terry, 1952) and BERT classifier (Devlin et al., 2019); see §4.2 in Ong et al. (2024).

**Remark 1** *Our empirical study suggests that rescaling $\{r_i\}_{i=1}^m$ by a normalization constant $c > 0$ to $\{c \cdot r_i\}_{i=1}^m$, so that the rescaled values are on the same scale as $\{y_i\}_{i=1}^n$, can substantially improve the performance of our proposed router. Some normalization constant could be considered include: (1) $c$ normalizing the magnitude: $\max\{|c \cdot r|_i\}_{i \in [n]} = \max\{|y|_i\}_{i \in [m]}$; (2) $c$ normalizing the empirical variance: $\mathrm{Var}(c \cdot r_i) = \mathrm{Var}(y_i)$; (3) $c$ (approximately) minimizing the distribution distance (e.g., 2-Wasserstein distance) between the empirical distributions of $\{c \cdot r_i\}_{i \in [n]}$ and $\{y_i\}_{i \in [m]}$.*

## 2.2 Cost function

For any LLM $\mathcal{M}$, we define its cost function as $\mathcal{C}_{\mathcal{M}} : \mathcal{Q} \mapsto \mathbb{R}_{>0}$ that quantifies the cost of generating the answer for any input query $q \in \mathcal{Q}$ using LLM model $\mathcal{M}$. Following others (Ong et al., 2024; Ding et al., 2024), in this paper, we assume the cost functions of both models are known a priori, and consider the following normalized cost functions:

$$\mathcal{C}_{\mathcal{M}_p}(q) = 1, \quad \mathcal{C}_{\mathcal{M}_a}(q) = 0, \tag{3}$$

for any $q \in \mathcal{Q}$. Such cost functions treat the call of $\mathcal{M}_p$ as one unit more expensive than the call of $\mathcal{M}_a$ for any query. We focus on this normalized cost mainly for the ease of exposition.

**Remark 2** *Our proposed method can be easily applied to more complicated and realistic cost functions. Many LLM providers (e.g., Claude, DeepSeek, Gemini and GPT) adopt a token-based pricing model for developers and enterprises, where the cost of a query is the sum of input tokens times the input rate and output tokens times the output rate (Chen et al., 2023). Formally, for LLM $\mathcal{M}$, $\mathcal{C}_{\mathcal{M}}(q) = c_{\mathrm{in},\mathcal{M}} \cdot \mathcal{T}_{\mathcal{M}}(q) + c_{\mathrm{out},\mathcal{M}} \cdot \mathcal{T}_{\mathcal{M}}(\mathcal{M}(q)) + c_{\mathrm{fix},\mathcal{M}}$, where $\mathcal{T}_{\mathcal{M}}(q)$ and $\mathcal{T}_{\mathcal{M}}(\mathcal{M}(q))$ are the input and output token counts, $c_{\mathrm{in},\mathcal{M}}, c_{\mathrm{out},\mathcal{M}}$ are known per-token rates, and $c_{\mathrm{fix},\mathcal{M}}$ is a fixed cost. Input tokens can be obtained via the tokenizer[1], while output tokens can be estimated using generation limits (OpenAI, 2024) or predictive methods (Zheng et al., 2023b). Latency may also be incorporated as an additional cost component.*

---

[1]*e.g.*, https://platform.openai.com/tokenizer

## 2.3 THE ROUTING DECISION RULE

The decision rule of an LLM router is designed to compare the quality gain of choosing $\mathcal{M}_p$ over $\mathcal{M}_a$ with the corresponding answer generation cost in §2.2. To quantitatively measure the quality gain of routing a new query $q$, previous works mainly leverage the average quality gain of different preference data $\eta(q)$ (Ong et al., 2024; Zhang et al., 2025). However, as we focus on fields requiring professional knowledge, *e.g.*, healthcare, science, and computer programming, the GS model $\psi(q)$ is arguably a more reliable measure of quality gain. Specifically, the proposed utility contrasts the expected quality gain based on the GS with the cost function and strives to balance between the response quality with the cost as follows:

$$\mathcal{D}(q \mid w, m) = \underbrace{\mathbb{E}\,(r \mid q)}_{\text{GS quality gain}} - w \cdot \underbrace{\left(C_{\mathcal{M}_p}(q) - \mathcal{C}_{\mathcal{M}_a}(q)\right)}_{\text{cost loss}} = \psi(q) - w \cdot \left(\mathcal{C}_{\mathcal{M}_p}(q) - \mathcal{C}_{\mathcal{M}_a}(q)\right). \quad (4)$$

Here, $w \geq 0$ is a user-specified conversion factor to control the *trade-off* between the quality gain and the additional cost if the expensive model $\mathcal{M}_p$ is preferred over $\mathcal{M}_a$. When $\psi(q)$ is known and cost function is binary as in (3), the Bayes optimal classifier selects $\mathcal{M}_p$ over $\mathcal{M}_a$ in response to the query $q$ if and only if the quality gain surpasses the required additional cost based on the decision rule, namely, $\psi(q) > w$, and selects $\mathcal{M}_a$ over $\mathcal{M}_p$ otherwise.

# 3 INTEGRATIVE LLM ROUTING THROUGH CAUSAL META-LEARNERS

## 3.1 ORACLE INTEGRATIVE ROUTER WITH KNOWN SHIFT FUNCTION

To efficiently evaluate the average quality gain function $\psi(\cdot)$ of the GS model, we aim to combine the information from both $\mathcal{D}_P$ and $\mathcal{D}_G$. However, due to the uncertainty of human and LLM judge's preference ratings, there may exist a potential discrepancy (bias) between the golden-labeled quality gain $\psi(\cdot)$ for $\mathcal{D}_G$ and the preference-choice model $\eta(\cdot)$ for $\mathcal{D}_P$ (Zheng et al., 2023a; Wataoka et al., 2024; Zhu et al., 2023; Szymanski et al., 2025). This bias can be quantitatively modeled as an unknown shift function for any query $q$,

$$\Delta(q) = \psi(q) - \eta(q).$$

Consequently, a regression approach using the directly combined data $\mathcal{D}_G \cup \mathcal{D}_P$ (Ong et al., 2024) can suffer from non-negligible estimation bias for $\psi(\cdot)$ even if the sample sizes of both PB data and the GS data are sufficient.

In this section, we estimate $\psi(\cdot)$ under an oracle scenario that the shift function $\Delta(\cdot)$ is *known* (a theoretical scenario for illustration purpose) and leave scenario of unknown $\Delta(\cdot)$ to section 3.3, where our new method developed. Under such an ideal condition, one can estimate $\eta(\cdot)$ by integrating the information in $\mathcal{D}_P$ and $\mathcal{D}_G$ using a bias correction process that takes the information of $\Delta(\cdot)$ into account. Specifically, consider the following bias-corrected human preference data:

$$\mathcal{T}(\mathcal{D}_P \mid \Delta) = \left\{(q_i', r_i' = y_i + \Delta(q_i'))\right\}_{i=1}^m,$$

where $r_i'$ can be roughly interpreted as the pseudo-GS quality difference as if the human-preference queries are prompted. Then, our newly enriched dataset after bias correction can be described as

$$\mathcal{D}^+ = \mathcal{D}_G \cup \mathcal{T}(\mathcal{D}_P \mid \Delta) = \{(q_i, r_i)\}_{i=1}^n \cup \{(q_i', r_i')\}_{i=1}^m.$$

Note that all samples in $\mathcal{D}^+$ are conditionally unbiased for $\psi(q)$, namely, for any $i \in [n]$ and $j \in [m]$,

$$\psi(q_i) = \mathbb{E}(r_i \mid q_i), \quad \psi(q_j') = \mathbb{E}(r_j' \mid q_j').$$

Over $\mathcal{D}^+$, one can apply any ML algorithm to estimate $\psi(\cdot)$ through a direct nonparametric regression. More specifically, $\psi(\cdot)$ solves the following population least-square problem:

$$\psi(\cdot) = \underset{h:\mathcal{Q}\mapsto\mathbb{R}}{\arg\min} \frac{1}{n+m}\mathbb{E}_{\mathcal{D}^+}\left(\sum_{(q,r)\in\mathcal{D}^+}(r - h(q))^2\right), \quad (5)$$

where the expectation is taken with respect to the distribution of $\mathcal{D}^+$. Here $h(\cdot)$ is an arbitrary prediction function mapping a query to a scalar estimation of the GS quality gain, and minimizing

(5) over all such $h(\cdot)$ identifies the true average GS quality gain $\psi(\cdot)$. Then, our oracle estimator is obtained by solving the (regularized) empirical counterpart of (5):

$$\hat{\psi}(\cdot \mid \Delta) = \underset{h \in \mathcal{H}_\Delta}{\arg\min} \frac{1}{n+m} \left[ \sum_{i=1}^{n} (r_i - h(q_i))^2 + \sum_{i=1}^{m} (\underbrace{y_i + \Delta(q_i')}_{r_i'} - h(q_i'))^2 \right] + \Lambda(h), \qquad (6)$$

where $\mathcal{H}_\Delta$ is the estimator class specified by the ML algorithm, *e.g.*, Gaussian process regression (Rasmussen & Williams, 2006), deep neural networks (Goodfellow et al., 2016), and random forests (Breiman, 2001a), and $\Lambda(\cdot)$ is an optional user-specified regularizer on the complexity of $h$, *e.g.*, the $\ell_2$ (ridge) regularizer (Tikhonov & Arsenin, 1977) and the $\ell_1$ (Lasso) regularizer (Tibshirani, 1996).

By appropriately choosing the ML algorithm (and hereby $\mathcal{H}_m$ in (6)), $\hat{\psi}(\cdot \mid \Delta)$ serves as a statistically principal estimator for $\psi(\cdot)$ using all samples in $\mathcal{D}_G \cup \mathcal{D}_P$. For example, if $\psi(\cdot)$ satisfies certain smoothness condition, then several nonparametric regression estimators can achieve statistical optimality; see *e.g.*, Wasserman (2006); Moutrada et al. (2020); Schmidt-Hieber (2020).

### 3.2 GS–PB DATA INTEGRATION: A CAUSAL INFERENCE PERSPECTIVE

In practice, the shift function $\Delta(\cdot)$ is unknown. Nevertheless, the oracle procedure outlined in §3.1 indicates that, empirically, it is crucial to develop a principal statistical estimation framework for the shift function $\Delta(\cdot)$ in order to estimate $\psi(\cdot)$ efficiently by combining the information from $\mathcal{D}_G$ and $\mathcal{D}_P$. In the following two sections, we reformulate the data integration problem under the potential outcome framework in causal inference (see *e.g.*, Imbens & Rubin (2015)), and correspondingly, $\Delta(\cdot)$ is the conditional average treatment effect (CATE) under such a new model formulation. One can then use well-developed CATE estimation approaches in causal inference, *e.g.*, meta-learners (Künzel et al., 2019), to estimate $\Delta(\cdot)$ robustly and efficiently.

To streamline the presentation, we pool the GS and PB datasets into a single collection and use a unified triple $(s_i, t_i, o_i)$ for sample $i$, where $s_i$ denotes the query of sample $i$, $t_i \in \{0, 1\}$ is the source indicator ($t_i = 1$ if the label is obtained from the gold-standard (GS) mechanism and $t_i = 0$ if it is obtained from the preference-based (PB) mechanism), $o_i$ is the observed outcome, i.e., the evaluated quality gain between $\mathcal{M}_p(s_i)$ and $\mathcal{M}_a(s_i)$ under the corresponding mechanism. With this notation, the pooled dataset $\mathcal{D}_G \cup \mathcal{D}_P$ can be written as

$$\mathcal{D} = \{(s_i, t_i, o_i)\}_{i=1}^{n+m}, \qquad (7)$$

where each sample comes from either $\mathcal{D}_G$ or $\mathcal{D}_P$ depending on $t_i$. Specifically, when $t_i = 1$ (GS sample), we have $o_i = r_i$ as in model 1; when $t_i = 0$ (PB sample), we have $o_i = y_i$ as in model 2.

Rather than modeling $\mathcal{D}_G$ and $\mathcal{D}_P$ separately, we can alternatively characterize the distribution of the combined dataset $\mathcal{D} = \mathcal{D}_G \cup \mathcal{D}_P$ using a hierarchical mixture model (Pooled DGP).

---

*Pooled Data Generation Process (Pooled DGP)*

*For each $(s_i, t_i, o_i) \in \mathcal{D}$:*

1. *Generate $t_i$ with $\Pr(t_i = 1) = \kappa \in [0, 1]$; here $\kappa$ controls how often GS samples are observed in the joint dataset.*
2. *Generate $s_i$ with $s_i \mid t_i = 1 \sim \mathcal{Q}$ and $s_i \mid t_i = 0 \sim \mathcal{Q}'$, where $\mathcal{Q}$ and $\mathcal{Q}'$ are the query distributions of the GS and PB data, respectively;*
3. *Generate $o_i = r_i$ under model (1) with $q_i = s_i$ if $t_i = 1$, and $o_i = y_i$ under model (2) with $q_i' = s_i$ if $t_i = 0$.*

---

Such a joint data generation process naturally leads to the causal potential outcome framework (Rubin, 2005). Specifically, we can view each query a unit, $s_i$ as its covariates, and consider $t_i \in \{0, 1\}$ as the binary treatment assignment to indicate whether the evaluation between $\mathcal{M}_p(s_i)$ and $\mathcal{M}_a(s_i)$ is carried out by gold standards ($t_i = 1$) or is PB ($t_i = 0$). For each query $s_i$, the two potential evaluation outcomes follow:

$$o_i^{(1)} = \psi(s_i) + \epsilon_i, \quad o_i^{(0)} = \eta(s_i) + \epsilon_i', \qquad (8)$$

where $o_i^{(1)}$ represents the counterfactual quality assessment of the quality gain shift from $\mathcal{M}_a(s_i)$ to $\mathcal{M}_p(s_i)$ if the evaluation is justified by the gold standards, while $o_i^{(0)}$ represents the quality gain with the same query, but the evaluation is judged through a preference-based mechanism. Then, samples in $\mathcal{D}$ can be equivalently considered as generated from the following standard causal mechanism.

**Lemma 1** *Define $f_{\mathcal{Q}}$ and $f_{\mathcal{Q}'}$ as density functions of $\mathcal{Q}$ and $\mathcal{Q}'$, respectively. Then the Pooled DGP is equivalent to the Causal DGP as follows.*

---

*Causal Data Generation Process (Causal DGP)*

*For each $(s_i, t_i, o_i) \in \mathcal{D}$:*

1. *Generate $s_i \sim \kappa\mathcal{Q} + (1 - \kappa)\mathcal{Q}'$, which is the mixture distribution of $\mathcal{Q}$ and $\mathcal{Q}'$ with the mixture proportion $\kappa$;*

2. *Generate $t_i$ following the propensity score model $\Pr(t_i = 1 \mid s_i) = p(s_i) := \kappa f_{\mathcal{Q}}(s_i)\{\kappa f_{\mathcal{Q}}(s_i) + (1 - \kappa)f_{\mathcal{Q}'}(s_i)\}^{-1}$;*

3. *Generate $o_i$ following the standard potential outcome model: $o_i = t_i o_i^{(1)} + (1 - t_i)o_i^{(0)}$, where $o_i^{(1)}$ and $o_i^{(0)}$ are given by (8).*

---

The proof of Lemma 1 is in Appendix A.5. Lemma 1 clarifies that the target function $\Delta(\cdot)$ is CATE from the perspective of causal data generation:

$$\Delta(s) = \psi(s) - \eta(s) = \mathbb{E}(o^{(1)} - o^{(0)} \mid s).$$

The causal identification assumptions such as consistency and unconfoundedness could be naturally satisfied under the Causal DGP. In particular, under the data collection procedure considered in this paper (c.f., §A.1), the no unmeasured confounders is satisfied, whenever there is no unobserved random variable, other than the query $s$, jointly affecting both the treatment assignment mechanism and the outcome. On the other hand, the positivity assumption on the propensity score, i.e., $p(s) \in (\epsilon, 1 - \epsilon)$ for some constant $\epsilon > 0$, may be violated when the supports of $\mathcal{Q}$ and $\mathcal{Q}'$ do not coincide. In particular, violation occurs if there exists a region of $q$ such that $f_{\mathcal{Q}}(s) > 0$ while $f_{\mathcal{Q}'}(s) = 0$, or vice versa. In such cases, our proposed method remains valid after a data truncation step: we estimate $\Delta(\cdot)$ only within the samples in the overlapped region of supports. We defer a detailed discussion of this truncation-based extension to future work in §5.

### 3.3 CAUSAL META-LEARNING FOR $\Delta(q)$ AND META-ROUTER

Building on the seminal work of Künzel et al. (2019), many causal meta-learning approaches are developed, aiming to provide principled and flexible frameworks for CATE estimation. Meta-learners can incorporate any off-the-shelf ML algorithm, thereby offering substantial flexibility. Moreover, by leveraging ideas from orthogonal ML and semiparametric statistics (see, *e.g.*, Chernozhukov et al., 2018), meta-learners such as the R-learner (Nie & Wager, 2021) and the DR-learner (Kennedy, 2023) enjoy the oracle property. In particular, under mild conditions of nuisance function estimation, CATE meta-learners can be asymptotically equivalent to an oracle estimator that has access to the full set of individual treatment effects $\{o_i^{(1)} - o_i^{(0)}\}_{i=1}^n$, whereas in practice only one of $o_i^{(1)}$ or $o_i^{(0)}$ is observed for each $i$. This oracle property implies that R-learner and DR-learner could achieve the statistical optimality for the estimation of $\Delta(\cdot)$ in our setting (Wu & Yang, 2022; Curth & Van der Schaar, 2021). In this paper, we focus on R- and DR-learners.

The implementation details of R- and DR-learners are deferred to §A.3 in the Appendix. Both learners offer robustness against nuisance model misspecification and fit naturally into our estimation purpose of $\Delta(\cdot)$. In this work, we consider both approaches as benchmark estimators for the shift function $\Delta(\cdot)$, and employ nonparametric ML regressors (*e.g.*, random forests, deep neural networks, and XGBoost) to capture heterogeneous structures of $\Delta(\cdot)$ across the query space.

The sample-splitting could be further employed into R- and DR-learners as discussed in (Nie & Wager, 2021; Kennedy, 2023) to avoid potential biases brought by nuisance function training through ML algorithms. We omit the details only for simplicity, and note that the sample splitting could be

straightforwardly incorporated into our method. We refer interested readers to the aforementioned two papers and, *e.g.*, Chernozhukov et al. (2018) for further discussions.

Building on the construction of the oracle router in (6), we now replace the known shift function $\Delta(\cdot)$ with its meta-learner-based estimator $\hat{\Delta}(\cdot)$, and thereby formalize our two-step meta-router.

---

**Meta-router**

Inputs: $\mathcal{D} = \mathcal{D}_G \cup \mathcal{D}_P$; $\mathcal{H}_\Delta$, $\mathcal{H}_m$, $\Lambda(\cdot)$ specified by selected ML algorithms.

1. Estimate the shift function $\hat{\Delta}(\cdot)$ via certain CATE learning approaches, *e.g.*, the R-learner or DR-learner in (S3) or (S4) with nuisance functions trained over $\mathcal{D}$.

2. Meta-router $\hat{\psi}(\cdot \mid \hat{\Delta})$ is obtained by solving (6) wherein $\Delta(\cdot)$ is replaced by $\hat{\Delta}(\cdot)$.

---

Although using DR-learner and R-learner as examples, our meta-router is a generally framework does not tie on any specific CATE estimation approach. Our meta-router framework could be naturally extended to the multiple-LLM scenario, we defer more discussions to §A.2 in the Appendix.

## 4 NUMERICAL EXPERIMENTS

### 4.1 HEALTHBENCH

HealthBench (Arora et al., 2025) is a recently released benchmark designed to evaluate the performances of LLMs in open-ended healthcare scenarios. It consists 5000 professional user-model dialogues that were selected to span a wide range of healthcare scenarios. In total, 262 physicians across 26 specialties and 60 countries contributed to the creation of evaluation rubrics and consensus standards, make the evaluation mechanism precise in reflecting the qualities of LLM responses. The meta-evaluation verifies the trustworthy of these rubrics in faithfully reflecting physician judgment.

In our numerical experiments, we set Gemini 2.5 Pro as the primary model $\mathcal{M}_p$ (Comanici et al., 2025) and Gemma 3 12B as the alternative model $\mathcal{M}_a$ (Team et al., 2025), and collect their responses to all HealthBench questions. We then employ GPT-5-mini (OpenAI, 2025) for evaluation. For gold-standard evaluations, each score-collecting prompt includes the evaluation rubrics, the original question, and the model response, and GPT-5-mini is asked to assign a score strictly following the official rubrics. Notably, generating GS evaluation through LLM could be a limitation of our study, and direct expert validation shall be ideal. The HealthBench study (OpenAI, 2025) reports that GPT-4.1 with rubric achieves marco F1 score of 0.709 against physician annotations on consensus criteria and be able to match expert grading (Table 6 in OpenAI (2025)). Thus, we believe that using GPT5-mini with rubric should perform similarly to expert grading for our study. The score difference between $\mathcal{M}_p$ and $\mathcal{M}_a$ for each question is treated as the GS quality differences of two models (see Appendix A.7 for our prompts). For preference-based evaluation, each prompt contains only the question and the two responses, and GPT-5-mini, asked to act as a medical expert, indicates whether $\mathcal{M}_p$ is better (1), comparable (0), or worse (−1), and this returned value is treated as the PB quality gain (see Appendix A.7 for our prompts). We normalize two types of quality gain evaluations to align their empirical variance (c.f., Remark 1(2)). We embedded each query text to a 768-dimensional vector using the gemini-embedding-001 model. We report in Figure S2 in Appendix, the histogram of the PB–GS quality differences $\{r_i - y_i\}_{i=1}^{5000}$. The sample mean of these differences is substantially below zero, as confirmed by a two-sided t-test yielding a p-value smaller than $2.2 \times 10^{-16}$, which motivates the training of debiased meta-router.

**Experiment Setting** For each Monte Carlo (MC) round, we specify the estimator class $\mathcal{H}$ by a machine learning algorithm, a GS sample size $n$, and a dimension $d$ such that we further reduce the dimension of query text embedding to $d$ via PCA; for simplicity, we use the same estimator class $\mathcal{H}$ for all nuisance function, CATE function and router training. We then randomly split the data into three parts: a testing set $\mathcal{D}_{\text{text}}$ of with 500 queries and the corresponding GS evaluation outcomes $r_i$, a GS training set of size $n$, and a PB training set containing the remaining samples. Each training set only includes its corresponding type of evaluation outcomes. We compare seven types of routers: (1) an oracle benchmark router that has access to the GS evaluation outcomes for all training queries in both GS and PB sets, and trains $\psi(q)$ over $\mathcal{H}$ using all these outcomes;

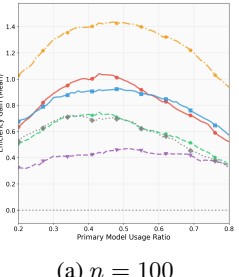 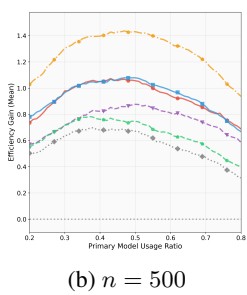 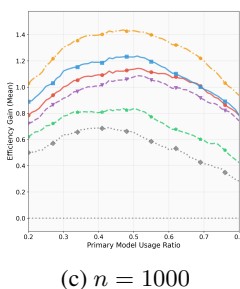

(a) $n = 100$                (b) $n = 500$                (c) $n = 1000$

Figure 1: The efficiency gains of different routing strategies compared to the random routing baseline, against the primary model usage ratio in the main numerical experiments. Subfigures correspond to varying GS sample sizes. Colors indicate different methods: oracle benchmark, meta-router via DR-learner, meta-router via R-learner, predictive router using pooled data, predictive router using GS data only, and predictive router using PS data only.

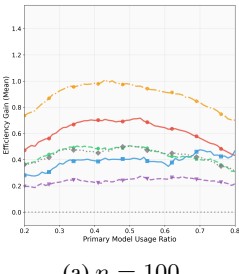 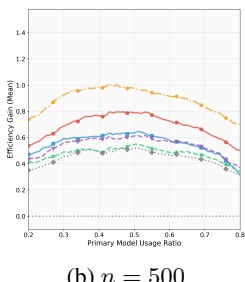 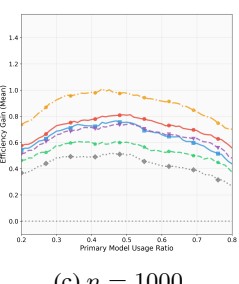

(a) $n = 100$                (b) $n = 500$                (c) $n = 1000$

Figure 2: The efficiency gains of different routing strategies compared with the random routing baseline versus the primary model usage ratio. All regressions are implemented via XGBoost. Other settings are the same as Figure 1.

(2) a predictive router that estimates $\psi(q)$ over $\mathcal{H}$ on the pooled GS and PB training data, without distinguishing evaluation types; (3) a predictive router that estimates $\psi(q)$ over $\mathcal{H}$ using only the PB training data; (4) a predictive router that estimates $\psi(q)$ over $\mathcal{H}$ using only the GS training data; (5) a meta-router based on the R-learner trained on GS and PB data, with all involved predictions run by $\mathcal{H}$; (6) a meta-router based on the DR-learner trained on GS and PB data, with all involved predictions run by $\mathcal{H}$; (7) a random router that assigns each query to $\mathcal{M}_p$ with a fixed assignment probability. The routers based on pooled GS and PB training data, and solely based on PB training data, follow the same framework as Ong et al. (2024), serving as our state-of-art baseline.

**Main Experiments** We specify $\mathcal{H}$ by the learning algorithm of random forest (Breiman, 2001b), set PCA dimension $d$ to 50, and test three GS sample sizes $n \in \{100, 500, 1000\}$. For each configuration and each Monte Carlo (MC) round, each router's decision rule follows (4), with $\psi(q)$ replaced by the corresponding estimator and binary cost functions as in (3). Given any weight $w$ in (4), we compute the total efficiency (TE) of each router as TE $= \sum_{(q_i, r_i) \in \mathcal{D}_{\text{test}}} \mathbb{I}\{q_i$ is assigned to the primary model$\} \times r_i$, where $r_i$ denotes the realized quality gain. By varying $w$, or equivalently the assignment probability for the random router, we obtain TE values under different primary model usage ratios (PMUR), defined as the proportion of queries assigned to the primary model among all testing samples. We run 200 MC rounds for each configuration and report the mean TE across rounds for each router and PMUR level. To quantify relative performance, we further calculate the efficiency gain (EG) of a router as its improvement over the random router, averaging over 500 test samples:

$$\text{EG of any router} = \frac{\text{Mean TE of any router} - \text{Mean TE of the random router}}{500}.$$

The EGs of different routers versus PMURs under different sample sizes, are reported in Figures 1. The illustrative code to reproduce Figures 1(a) is available at https://github.com/yichistat/Meta-router.

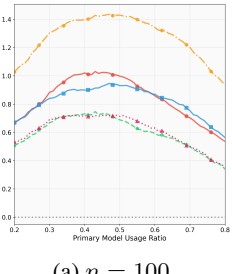 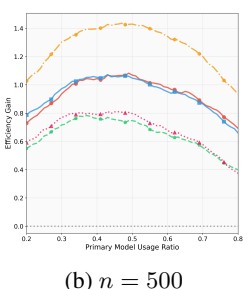 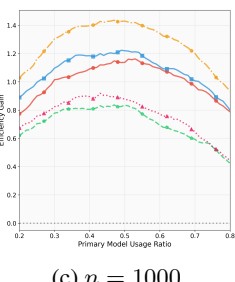

(a) $n = 100$          (b) $n = 500$          (c) $n = 1000$

Figure 3: The efficiency gains of different routing strategies compared with the random routing baseline versus the primary model usage ratio. The setting is same as Figure 1, with an additional curve corresponding to the simple debiased router through linear scaling.

Our simulation results demonstrate the superior efficiency of meta-routers, particularly in imbalanced regimes with very limited GS data. In contrast, the predictive router trained on directly pooled GS and PB data or only PB data, as considered in *e.g.*, Ong et al. (2024), shows little efficiency improvement even with relatively large GS sample sizes, highlighting the detrimental effect of bias $\Delta(q)$ in LLM routing. As the GS sample size increases, the efficiency gains of all routers improve, except for the PB-only router, highlighting the value of incorporating GS data for debiasing.

**Ablation Studies** To investigate the impacts of different meta-router components, we conduct different numerical experiments for ablation studies, by changing one key element in our main numerical experiments while keeping other settings unchanged, examine the performance changes.

(i) We consider $\mathcal{H}$ to be specified by another machine learning algorithm XGBoost (Chen & Guestrin, 2016). The EGs are reported in Figure 2. The R-learner-based router consistently outperform other routers, which demonstrates its robustness over different regression methods. In general, the EGs in Figure 2 are not as high as the EGs in Figure 1, showing the importance of comparing different regression methods when training the meta-router.

(ii) We consider another router trained using a simple debiasing strategy based on linear scaling. We debias the PB data by subtracting the sample mean difference between the PB and GS datasets, and then train a router on the pooled set consisting of the shifted PB data and the GS data. The resulting EGs are shown in Figure 3. The simple debiasing router has a similar performance with the router trained by directly pooled data, and Meta-Routers consistently outperform it. Such observation indicates that in practice, the bias between PB and GS outcomes are usually heterogeneous across different queries, and more sophisticated CATE estimators such as metalearners, are essential for an effective debiasing.

(iii) In the data preprocess, we do not normalize the GS data following Remark 1(2). The EGs are reported in Figure S4. The meta-routers do not significantly outperform the router trained via GS data only, highlighting the importance of pre-normalization for meta-routers.

(iv) We consider $d = 100$ for the PCA. The EGs are reported in Figure S3. The meta-routers also outperform other routers, further demonstrating the robustness of our approach.

(v) We collect the PB data alternatively from another cheaper LLM judge: Grok 4 Fast, with other settings kept the same as the main numerical experiment. The EGs are reported in Figure S5. The meta-routers outperform other approaches especially when the sample size of GS data is small, which verifies the adaptivity of our approach with different preferences.

## 4.2 PRBENCH

PRBench is a rubric-based benchmark for high-stakes professional reasoning in the domain of law and finance (Akyrek et al., 2025). The dataset comprises 1,100 expert-authored tasks across 114 countries and 47 U.S. jurisdictions, similar to HealthBench, accompanied by 19,356 expert-curated evaluation rubrics. All tasks were contributed by 182 industry professionals, thus offers rich real-world complexity beyond conventional academic benchmarks, enabling deeper analysis of open-ended, economically consequential reasoning. We focus on 676 questions in PRBench with one-turn conversation. Similar to §4.1, we consider the primary model as Gemini 2.5 Pro and alterna-

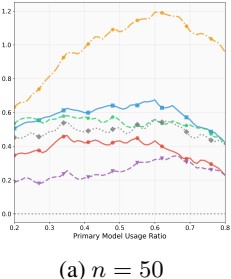 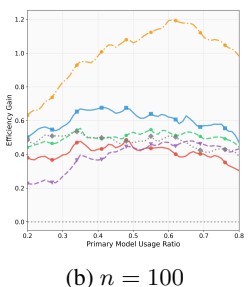 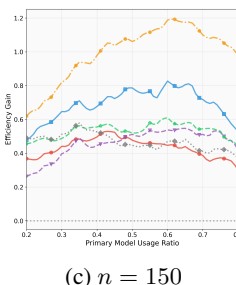

(a) $n = 50$      (b) $n = 100$      (c) $n = 150$

Figure 4: The efficiency gains of different routing strategies trained and tested over PRBench in §4.2. Explanations of subfigures are the same as Figure 1.

tive model Gemma 3 12 B, and use the same mechanism for the GS-based and PB-based answer evaluations; see Appendix A.7 for our prompts. Due to the limited sample size of PRBench, we correspondingly consider small GS sample size $n \in \{50, 100, 150\}$ and small PCA dimension for the query embeddings, namely, $d = 20$. Other numerical experiment settings are the same as the main experiment in §4.1.

The EGs are reported in Figure 4. The DR-learner-based router consistently outperforms the baselines, demonstrating the effectiveness of our approach. Under the limited sample size, the R-learner-based router offers no clear advantage over other methods, highlighting the superior sample efficiency of the DR-learner in this setting.

## 5   FUTURE WORK: TRUNCATION-BASED META-ROUTER UNDER POSITIVITY VIOLATION

Currently, our framework requires that the query distribution of GS data and that of the PB data share the common support, i.e., the positivity of propensity scores shall hold. This requirement can be violated in practice when, *e.g.*, the GS data focuses on one category where responses can be easily justified, while the PB data are with regard to more subjective queries. One could avoid positivity violation by explicit experiment designs in the data collecting period. On the other hand, positivity violation could also be detected through high-dimensional density ratio estimation of the query distributions in GS and PB data, respectively; see *e.g.*, Sugiyama et al. (2012). In particular, the region where the estimated density ratio is well upper- and lower-bounded can be interpreted as the overlap region between the distributions of $q$ in the GS and PB datasets, respectively.

When the propensity scores tend to be extreme (i.e., close to 0 or 1), the R-learner and EP-learner (van der Laan et al., 2024) may offer more robust debiasing performance. When the positivity assumption is totally violated, the distribution supports of two query distributions do not fully overlap. A promising direction is to develop a truncation-based meta-router, which always incorporates all GS data but only retains preference data within the estimated overlap region of the two distributions. In particular, the overlap can be identified via efficient density ratio estimation. Then a meta-learner of $\Delta(\cdot)$ is trained only through overlapping samples in $\mathcal{D}_G \cup \mathcal{D}_P$, which are considered as belonging to this region. If following Remark 1(2), GS data are now only normalized to have the same variance as PB data *within* these overlapping samples. Finally, when we train our truncation-based meta-router by solving (6) with obtained $\hat{\Delta}(\cdot)$ but only incorporating the samples $\mathcal{D}_P$ which belong to the detected overlap region. This truncation-based strategy offers a principled way to exploit abundant preference data while avoiding extrapolation bias outside the common support.

Additional discussions on other future directions are included in Appendix A.4, including the applications of semi-supervised learning and active learning, and the potential extension to out-of-distribution routing.

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

# A    APPENDIX

## A.1    PRACTICAL DEPLOYMENT AND END-TO-END WORKFLOW

In this section, we discuss design choices and recommendations for deploying meta-router in an end-to-end workflow. To make this section self-contained, we briefly recall the key parameters and notations. Following the main paper, we consider pairwise LLM routing between two models: a higher-quality, higher-cost model $\mathcal{M}_p$ (the "primary") and a cheaper alternative $\mathcal{M}_a$. In practice, $\mathcal{M}_p$ would typically be the strongest model available in the stack (for example, a proprietary frontier LLM), and $\mathcal{M}_a$ a smaller or open-source model chosen for lower price or latency. A router such as meta-router observes an incoming query $q_i \in \mathcal{Q}$ and decides whether to assign it to $\mathcal{M}_p$ or $\mathcal{M}_a$ based on the *expected gold-standard quality gain* $\psi(q_i)$ and the associated cost, following the decision rule in Section 2.3.

The first design choice is the gold-standard quality objective. In practice, the operator must choose the evaluation mechanism that defines the evaluated quality gain $r_i$ between $\mathcal{M}_p(q_i)$ and $\mathcal{M}_a(q_i)$ and hence the gold-standard quality gain function $\psi(\cdot)$. For example, when correctness is objectively defined, $r_i$ can be a discrete gain such as 1, 0, or $-1$ depending on which model answers correctly. When evaluation is rubric-based, domain experts (or a trusted evaluation pipeline) score the two responses separately and $r_i$ is defined as the difference between these scores. In other applications, an internal reward model may provide a scalar for each response, and $r_i$ can again be taken as the difference between the reward assigned to $\mathcal{M}_p(q_i)$ and $\mathcal{M}_a(q_i)$. Meta-router does not require access to per-response scores beyond this scalar difference. We recommend that operators define the quality gain using an evaluation mechanism that is as objective, stable, and aligned with the target task as possible, since the router is explicitly optimized for this gold-standard objective.

The second design choice is the cost model and the acceptable trade-off between quality and cost, which are encoded through a conversion factor $w$ (see Section 2.2). In deployment, the per-query cost of $\mathcal{M}_p$ and $\mathcal{M}_a$ can be measured in monetary units (for example, token-based API pricing), latency, or a weighted combination of the two. Given these costs, $w \geq 0$ controls how much gold-standard quality gain is required to justify the additional cost of using $\mathcal{M}_p$ instead of $\mathcal{M}_a$: larger values of $w$ favor cheaper routing, whereas smaller values favor higher-quality routing. When $\psi(q_i)$ is known, the Bayes-optimal policy routes $q_i$ to $\mathcal{M}_p$ if and only if $\psi(q_i)$ exceeds the cost-adjusted threshold implied by $w$ (Section 2.3). In practice, Meta-router learns an estimate $\hat{m}(q_i)$ and applies the same threshold rule. A practical way to select $w$ is to evaluate, on a held-out set with gold-standard labels, the average realized cost and average gold-standard quality achieved by the induced routing policy over a grid of candidate $w$ values, and then choose the smallest $w$ that satisfies a deployment budget constraint such as a maximum fraction of queries routed to $\mathcal{M}_p$ or a maximum total cost relative to an "always $\mathcal{M}_p$" baseline.

The third design choice is how to collect the datasets needed to train the router. Meta-router uses two data sources: a small gold-standard set $\mathcal{D}_G$ and a larger preference-based set $\mathcal{D}_P$. $\mathcal{D}_G$ is constructed by sampling queries from the actual traffic in the domain of interest and obtaining $r_i$ for each, via expert annotation or a trusted evaluation pipeline as discussed above. The sample size can be adapted to resources; in our experiments, a few hundred gold-standard queries already provide measurable gains. In parallel, $\mathcal{D}_P$ is obtained for queries drawn from the same traffic by collecting cheaper pairwise judgments (for example, crowdsourced labels or LLM-as-a-judge comparisons) indicating whether $\mathcal{M}_p$ is better, similar, or worse than $\mathcal{M}_a$. These judgments are coded as $y_i \in \{-1, 0, 1\}$. The important practical point is that gold-standard data can be scarce, expensive, and domain-specific, whereas preference data can be plentiful but biased. Meta-router is specifically designed to combine these two data resources and to correct the systematic bias in $\mathcal{D}_P$ using the information in $\mathcal{D}_G$.

The fourth design choice is the representation used for training and for incoming queries. In a deployed system, it is natural to reuse an existing embedding service (for example, the same text embedding model already used for retrieval). Each query in $\mathcal{D}_G \cup \mathcal{D}_P$ is embedded as a numerical vector, once using this service, and the resulting vectors (optionally reduced in dimension by principal component analysis) serve as features for all downstream components in meta-router. Leveraging existing infrastructure makes the proposed meta-router highly efficient and flexible, as the additional computational cost at training and inference time is dominated by a single embedding call and lightweight tabular models.

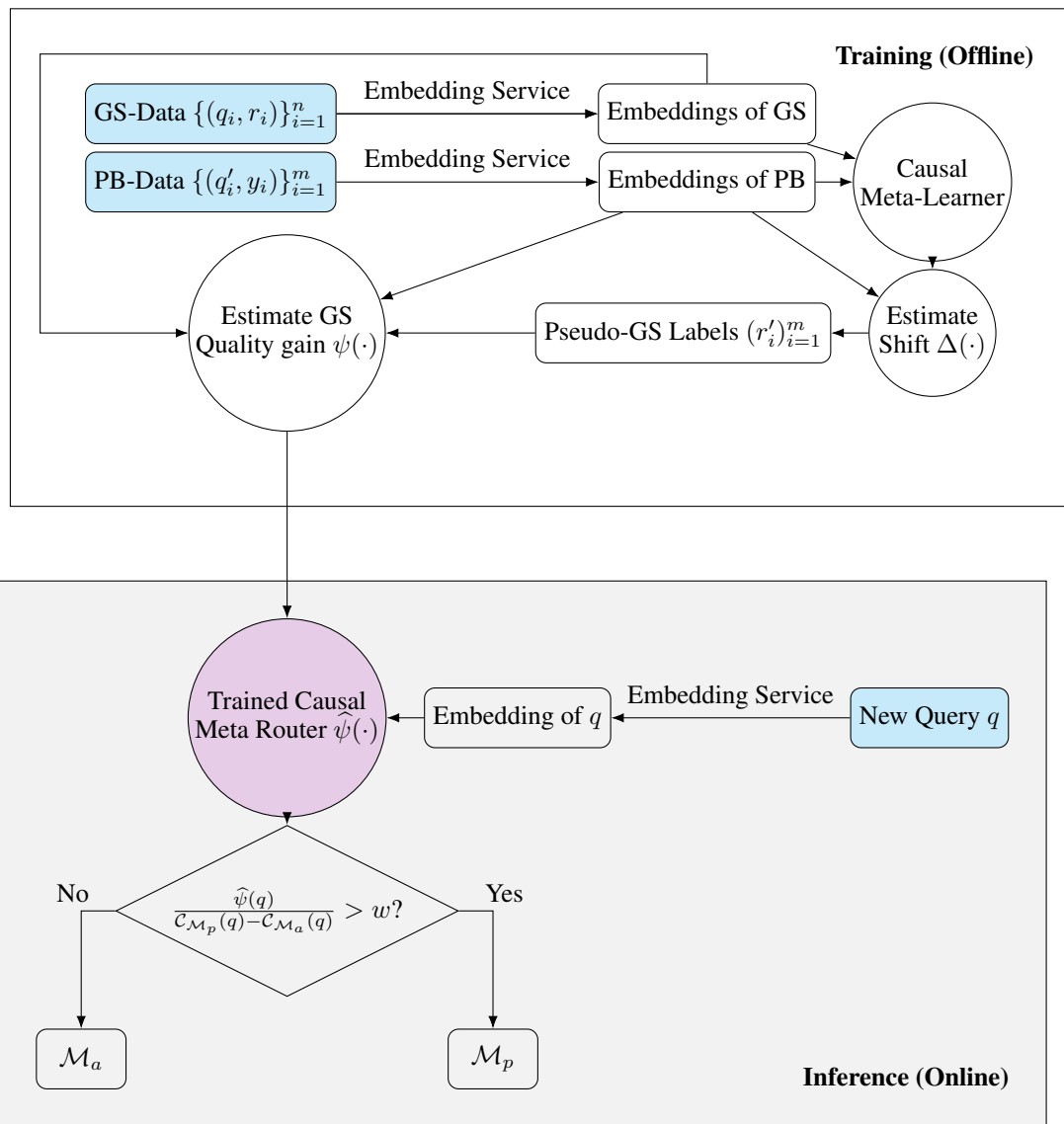

Figure S1: End-to-End workflow of the meta-router. The training stage only involves the GS-data $\{(q_i, r_i)\}_{i=1}^n$ and PB-data $\{(q_i', y_i)\}_{i=1}^m$ and can be carried out completely offline. The inference stage is based on the trained causal meta router $\widehat{\psi}(\cdot)$ and runs online with generic incoming new queries.

Given these design choices, meta-router is trained as described in Section 3.2: it learns the query-dependent shift between gold-standard and preference-based evaluations using causal meta-learners, uses this estimated shift to transform preference labels into pseudo-gold-standard labels, and then fits a final regression model $\hat{\psi}(q)$ on the union of true and pseudo gold-standard labels. We recommend using simple tabular learners such as gradient-boosted trees or random forests on the fixed embeddings, since these models are already powerful enough for the routing task while keeping computational cost minimal at inference time.

At inference time, the router is straightforward to integrate into existing infrastructure. Each incoming query from any supported domain is sent to the embedding service, the embedding is passed through the trained router $\hat{\psi}$, and the output is compared to the threshold $w$. If $\hat{\psi}(q) > w$, the query is routed to $\mathcal{M}_p$; otherwise it is routed to $\mathcal{M}_a$. The incremental latency compared with a system without routing is limited to one embedding call and a single evaluation of a small regressor, which

is negligible relative to executing the primary LLM. The entire workflow of the training and inference stages is visualized in Figure S1. Note that the training stage can be done offline with already available GS and PB data, while the trained causal meta-router can be directly applied online with incoming new queries.

Two additional considerations may arise in practice. The first is how to handle multiple or evolving domains. When domains are clearly distinct (for example, medical, legal, and coding assistance), the operator may train either a separate router for each domain or a single router that takes a domain indicator as an additional feature. In the latter case, the causal shift $\Delta(q)$ is allowed to vary by domain, and the router learns to use gold-standard supervision from one domain to inform others only to the extent that queries are similar in the shared embedding space. When a completely new domain is introduced, the recommended procedure is to start with preference-based data in that domain, then gradually collect a small amount of gold-standard data and retrain or fine-tune the router, exactly as in the initial deployment. Our experiments show that a modest number of domain-specific gold-standard queries is sufficient to obtain benefits from meta-router; without any gold-standard data in a domain, no current method can align routing decisions with that domain's gold-standard objective due to the underlying bias between gold-standard data and preference-based data.

The second consideration is monitoring costs and benefits after deployment. Because the router's objective is defined in terms of $\psi(q)$ and cost, it is natural to monitor, on a rolling basis, (i) the fraction of queries sent to $\mathcal{M}_p$, (ii) the realized cost relative to baselines such as always using $\mathcal{M}_p$ or always using $\mathcal{M}_a$, and (iii) the realized gold-standard quality on a small stream of queries that continue to receive expert evaluation. If the observed cost is too high, the operator can increase $w$; if the observed quality is lower than desired, the operator can decrease $w$ or collect additional gold-standard labels and retrain. Because router retraining is cheap, these updates can be performed regularly as query distributions or cost constraints change.

## A.2 MULTI-MODEL META-ROUTER

We discuss the natural extension of our meta-routing algorithm to the multi-model routing scenario. In particular, we attempt to route each query over $N$ candidate LLMs, indexed by 1 through $N$. Following the definition of the pooled dataset for two specific LLMs in (7), we use a quintet $(s_i, k_i, \ell_i, t_i, o_i)$, to define the $i$th collected pairwise comparison sample where the interpretations of $s_i, t_i, o_i$ are the same as the two-LLM routing scenario, i.e., they are the testing query, GS–PB indicator, and the quality gain measurement, respectively. The new variables $k_i, \ell_k \in [N]$ represent that the pair of LLMs being compared are LLM $k_i$ and LLM $\ell_i$ in the $i$th sample; without loss of generality, we require $k_i > \ell_i$ for all $i \in [N]$. The overall pairwise comparison dataset, comparing different pairs of LLMs, is

$$\mathcal{D} = \{(s_i, k_i, \ell_i, t_i, o_i)\}_{i=1}^{I},$$

where $I$ represents the full sample size.

We treat the query $s$ and the LLM pair $(k, \ell)$ together as the covariates, $t$ as the treatment assignment and $o$ as the observed outcome, for our causal framework. Then following the potential outcome framework in (8), we define the potential outcomes for the $i$th sample as

$$o_i^{(1)} = \psi_{k_i, \ell_i}(s_i) + \epsilon_i, \qquad o_i^{(0)} = \eta_{k_i, \ell_i}(s_i) + \epsilon_i', \tag{S1}$$

where the nuisance functions $\psi_{k, \ell}(s)$ and $\eta_{k, \ell}(s)$ now depend on both query $s_i$ as well as the LLM pair $(k, \ell)$ being compared. They represent the expected quality difference between the two models (LLM $k$ and LLM $\ell$) when assessed through the gold-standard evaluation and the human-preference evaluation, respectively. Then the bais between $\psi_{k, \ell}(s)$ and $\eta_{k, \ell}(s)$ could still be viewed as the CATE function:

$$\Delta_{k, \ell}(s) = \psi_{k, \ell}(s) - \eta_{k, \ell}(s) = \mathbb{E}\left(o^{(1)} - o^{(0)} \mid s, g = (k, \ell)\right).$$

Therefore, by treating $(s, k, \ell)$ instead of $s$ as the covariates, the meta-learners in §3.3 could still be exploited to obtain an estimator $\hat{\Delta}_{k, \ell}(s)$ of $\Delta_{k, \ell}(s)$ for any $(k, \ell, s)$. Then similar to (6), the meta-router could be obtained by solving the debiased empirical least-square objective,

$$\hat{\psi}_\star(\cdot \mid \hat{\Delta}) = \underset{h_\star(\cdot) \in \tilde{\mathcal{H}}_\Delta}{\arg\min} \frac{1}{I} \left[ \sum_{t_i=1} (o_i - h_{k_i, \ell_i}(s_i))^2 + \sum_{t_i=0} (o_i + \hat{\Delta}_{k_i, \ell_i}(s_i) - h_{k_i, \ell_i}(s_i))^2 \right] + \Lambda(h), \tag{S2}$$

where the trained router $\hat{\psi}_{k,\ell}(s \mid \hat{\Delta})$ now depends on both the query $s$ as well as the LLM pair $(k, \ell)$, and $\tilde{\mathcal{H}}_\Delta$ is any user specified estimator class containing functions approximating $\Delta_\star(\cdot)$ depending on both the LLM pair and query.

When estimating $\psi_\star(\cdot), \eta_\star(\cdot)$ and $\Delta_\star(\cdot)$, additional structural assumptions on these functions could be further made. For example, if considering ranking models like Bradley-Terry-Luce Model (Bradley & Terry, 1952; Luce et al., 1959) for the PB data generation (Rafailov et al., 2023), we have

$$\eta_{k,\ell}(s) = \frac{\exp(\theta_k(s))}{\exp(\theta_k(s)) + \exp(\theta_\ell(s))},$$

where $\theta_k(s)$ is the preference score function for each LLM $k$. Such modeling resolves the non-identification issue of $\eta_{k,\ell}(s)$ if $(k, \ell)_{k>\ell}$ does not get compared in $\mathcal{D}$, and reduce the sample complexity for the estimation of $\eta_\star(\cdot)$. For the practical implementation of meta-router with multiple LLMs in the above procedure, it would be important to investigate reasonable functional assumptions in order to improve the estimation flexibility and efficiency, which we leave for future work.

## A.3    R-LEARNER AND DR-LEARNER

**R-learner**    Let $\gamma(s) = \mathbb{E}(o \mid s)$ denote the marginal regression of the evaluation outcome on the covariates (query) $s$, and let $p(s) = \Pr(t = 1 \mid s)$ denote the propensity score of receiving a GS evaluation. R-learner (Nie & Wager, 2021) constructs the orthogonalized residuals:

$$\tilde{o}_i = o_i - \hat{\gamma}(s_i), \quad \tilde{t}_i = t_i - \hat{p}(s_i),$$

where $\hat{\gamma}$ and $\hat{p}$ are any sensible sample-based estimators for $\gamma$ and $p$. The R-learner then estimates $\Delta(\cdot)$ by solving the generalized least squares problem

$$\widehat{\Delta}_R(\cdot) = \underset{h \in \mathcal{H}_\Delta}{\arg\min} \frac{1}{n+m} \sum_{i=1}^{n+m} \left( \tilde{o}_i - \tilde{t}_i h(s_i) \right)^2 + \Lambda(h), \tag{S3}$$

where $\mathcal{H}_\Delta$ is a pre-specified hypothesis space (*e.g.*, linear functions, random forests, or neural networks), and $\Lambda(h)$ is a regularizer to control complexity. This formulation is quasi-oracle efficient under mild conditions on nuisance estimators. Specifically, causal forests (Athey et al., 2019) is associated with the tree-based function class $\mathcal{H}_\Delta$ that can flexibly capture heterogeneous structures of $\Delta(\cdot)$ across different $q$.

**DR-learner**    An alternative is the doubly robust (DR) learner of Kennedy (2023). It constructs a pseudo-outcome for each sample by combining outcome regression and propensity adjustment, thereby guaranteeing consistency if either component is correctly specified. Specifically, DR-learner considers $\mu_t(s) = \mathbb{E}(o \mid s, t)$, denoting the conditional regression under treatment status $t \in \{0, 1\}$. With no unmeasured confounders, we further have $\mu_1(\cdot) = \psi(\cdot)$, $\mu_0(\cdot) = \eta(\cdot)$. Then, the DR pseudo-outcome is

$$\tilde{\phi}_i = \left( \frac{t_i - \hat{p}(s_i)}{\hat{p}(s_i)(1 - \hat{p}(s_i))} \right) \left( o_i - \hat{\mu}_{t_i}(s_i) \right) + \hat{\mu}_1(s_i) - \hat{\mu}_0(s_i).$$

The DR-learner estimates $\Delta(\cdot)$ by regressing $\phi_i$ on $s_i$:

$$\widehat{\Delta}_{DR}(\cdot) = \underset{h \in \mathcal{H}_\Delta}{\arg\min} \frac{1}{n+m} \sum_{i=1}^{n+m} \left( \tilde{\phi}_i - h(s_i) \right)^2 + \Lambda(h). \tag{S4}$$

The doubly robust property ensures that $\widehat{\Delta}_{DR}(\cdot)$ is consistent if either $\mu_t(\cdot)$ or $p(\cdot)$ is estimated consistently. Such a feature is particularly appealing in our setting, because the distributional discrepancy between $\mathcal{D}_G$ and $\mathcal{D}_P$ may induce misspecification in one nuisance model.

**Remark 3 (Computational Cost)** *In practice, the computational cost of meta-learners is modest. The overall complexity is essentially the same order as training the underlying machine-learning models used within the learner. More concretely, the computation consists of: (i) fitting the nuisance models (propensity score and outcome regressions) and the final CATE regression, and (ii)*

*for certain meta-learners, constructing pseudo-outcomes. Step (i) has the same computational order as training the chosen ML algorithm for nuisance and CATE function approximations. Step (ii) requires only a single pass through the data (e.g., computing R-learner or DR-learner pseudo-outcomes), which is linear in the sample size. Therefore, the additional overhead introduced by meta-learning is mild relative to the ML models used.*

## A.4 FUTURE DIRECTIONS

**Semi-supervised learning** From a causal perspective, fully semi-supervised CATE estimation is technically challenging because the target is a high-dimensional function of the covariates; most existing semi-supervised learning based work (Cheng et al., 2021; Hou et al., 2025) focused on average treatment effects (ATEs) rather than CATEs. That said, we believe unlabeled data can still be very useful in our setting by helping to learn better query representations. One natural extension is to augment the current methods with a learnable representation that is trained on both labeled and unlabeled queries. Unlabeled queries from real traffic can regularize so that it reflects the true deployment distribution (e.g., via smoothness/consistency or clustering objectives), while GS+PB queries drive the CATE loss in this learned space. We believe that such representation can reduce the distribution difference between GS, PB, and incoming queries, thereby improving and downstream routing quality.

**Active learning** Active learning offers a complementary and appealing extension (Settles, 2009). In particular, rather than treating the GS pool as fixed, one could use an initial Meta-Router to adaptively select which queries receive expensive GS evaluation to maximize routing accuracy within a fixed GS budget. For example, one can view the evaluation mechanism (GS vs PB) as treatment and design acquisition rules that prioritize queries where the current router is most uncertain or most decision critical, such as queries near the routing decision boundary.

**Handling out-of-distribution routing** Our current work focuses on the in-distribution setting, where deployment queries are drawn from the same population as the GS and PB data used to train Meta-Router. For truly out-of-distribution (OOD) queries, a practical platform may collect responses from both models and obtain PB or GS evaluations for these new queries. This naturally forms an online-learning process in which the system gradually expands the coverage of the in-distribution domain. Integrating such OOD-aware data collection into Meta-Router is an interesting direction for future work, and we have noted this in the revised manuscript.

## A.5 PROOF OF LEMMA 1

The density function of $(s, t, o)$ in GS–PB DGP could be written as

$$f(s, t, o) = \kappa^t (1 - \kappa)^{1-t} f_{\mathscr{Q}}^t(s) f_{\mathscr{Q}'}^{1-t}(s) f_r^t(o \mid s) f_y^{1-t}(o \mid s),$$

where $f_r(\cdot \mid s)$ and $f_y(\cdot \mid s)$ represent the conditional probability density function of $r_i$ and $y_i$ given $q_i = s$, following (1) and (2), respectively. This could be further written as

$$f(s, t, o) = \underbrace{(\kappa f_{\mathscr{Q}}(s) + (1 - \kappa) f_{\mathscr{Q}'}(s))}_{f_{\kappa \mathscr{Q} + (1-\kappa) \mathscr{Q}'}(s)} \cdot \underbrace{\frac{\kappa^t (1 - \kappa)^{1-t} f_{\mathscr{Q}}^t(s) f_{\mathscr{Q}'}^{1-t}(s)}{\kappa f_{\mathscr{Q}}(s) + (1 - \kappa) f_{\mathscr{Q}'}(s)}}_{Pr(t_i = t \mid s) = t p(s) + (1-t) p(s)} \cdot \underbrace{f_r^t(o \mid s) f_y^{1-t}(o \mid s)}_{f_{o(t)}(o \mid s)}, \quad \text{(S5)}$$

recalling the notation in Causal DGP, and thereby show the distributional equivalence of two processes. □

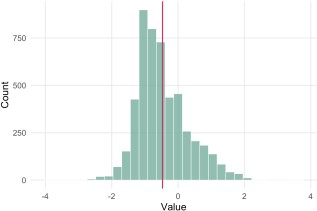

Figure S2: The histogram of all queries' PB-based and GS-based evaluation differences, i.e., $\{r_i - y_i\}_{i=1}^{5000}$ of HealthBench evaluations. The red vertical line represents the sample mean around $-0.47$.

### A.6 ADDITIONAL NUMERICAL RESULTS FOR §4.1

Additional results for §4.1 are reported in Figures S2-S5.

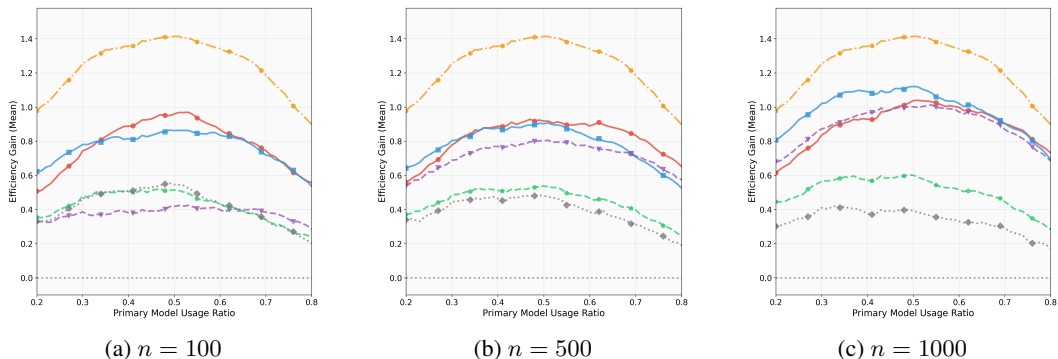

(a) $n = 100$  (b) $n = 500$  (c) $n = 1000$

Figure S3: The efficiency gains of different routing strategies compared with the random routing baseline versus the primary model usage ratio. The query embedding dimension is reduced to 100. Other explanations are the same as Figure 1.

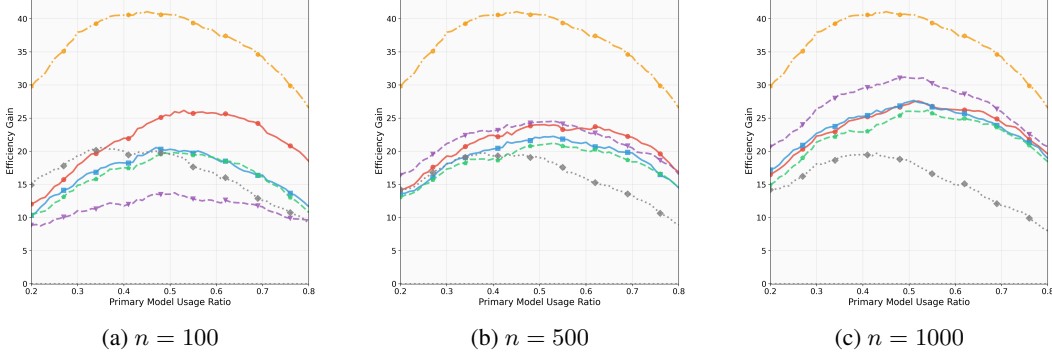

(a) $n = 100$  (b) $n = 500$  (c) $n = 1000$

Figure S4: The efficiency gains of different routing strategies compared with the random routing baseline versus the primary model usage ratio. GS data are not normalized to align the variance with PB data. Other settings are the same as Figure 1.

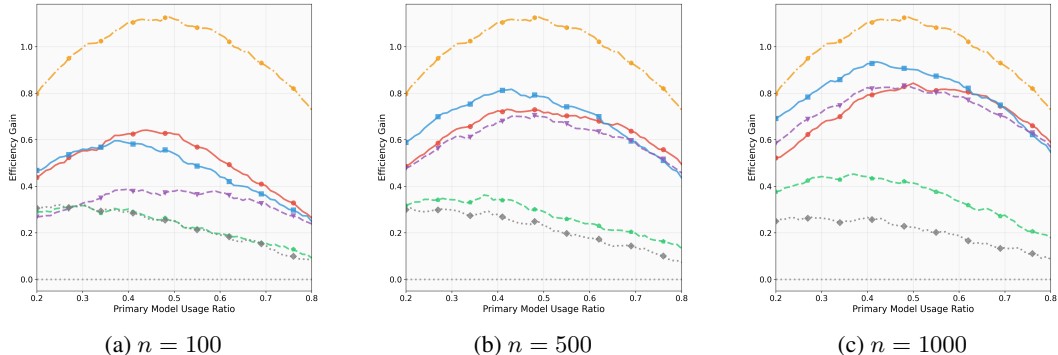

(a) $n = 100$    (b) $n = 500$    (c) $n = 1000$

Figure S5: The efficiency gains of different routing strategies compared with the random routing baseline versus the primary model usage ratio. PB data are collected by another LLM judger: Grok 4 Fast. Other settings are the same as Figure 1.

## A.7 EVALUATION-COLLECTING PROMPTS IN NUMERICAL EXPERIMENTS

In this section, we present the four prompts we use to collect the GS and PB evaluation data for our numerical experiments in §4.1 and §4.2 as follows.

---

**Prompt collecting the gold-standard evaluation from GPT-5 mini for HealthBench**

You are an expert medical educator and response evaluator. Evaluate this single medical response according to the provided criteria.

**Original Question:**
{prompt}

**Evaluation Criteria:**
{rubrics}

**Ideal Completion (Reference):**
{ideal_completions_data}

**Reponse to Evaluate:**
{response}

**Evaluation Rules:**

1. Read each criterion exactly as written

2. Award points only if the criterion is completely satisfied

3. Award 0 points for partial matches or close approximations

4. If a criterion specifies exact details (numbers, procedures), require exact matches

5. Be precise with medical/technical details

6. When in doubt, award 0 points rather than partial credit

7. For negative criteria (penalties), apply the full penalty if violated

**Output Format:**

```
{
    "total_score": <sum of all criterion points>,
    "criterion_scores": {
        "criterion_1": <points awarded/deducted>,
        "criterion_2": <points awarded/deducted>,
        "criterion_3": <points awarded/deducted>,
        ...
    }
}
```

---

**Prompt collecting the preference-based evaluation from GPT-5 mini for HealthBench**

You are a medical expert evaluating AI responses to healthcare questions. Your task is to compare two AI responses and determine which one is significantly better from a medical perspective.

**Evaluation Criteria:**

- Medical accuracy and safety

- Completeness of advice

- Appropriateness for the patient's situation

- Professional tone and clarity

- Adherence to medical best practices

**Instructions:**

1. Read the patient's question carefully

2. Evaluate both Response A and Response B

3. Return ONLY a single number:

   - 1 if Response A is significantly better
   - -1 if Response B is significantly better
   - 0 if both responses are roughly equivalent in quality

**Patient Question:**
{prompt}

**Response A:**
{response_a}

**Response B:**
{response_b}

**Your evaluation (return only 1, -1, or 0):**

---

## Prompt collecting the gold-standard evaluation from GPT-5 mini for PRBench

You are an expert legal and finance educator and response evaluator. Evaluate this single legal or finance response according to the provided criteria.

**Original Question:**
{prompt}

**Evaluation Criteria:**
{rubrics}

**Reponse to Evaluate:**
{response}

**Evaluation Rules:**

1. Read each criterion exactly as written

2. Award points only if the criterion is completely satisfied

3. Award 0 points for partial matches or close approximations

4. If a criterion specifies exact details (numbers, procedures), require exact matches

5. Be precise with medical/technical details

6. When in doubt, award 0 points rather than partial credit

7. For negative criteria (penalties), apply the full penalty if violated

**Output Format:**

```
{
    "total_score": <sum of all criterion points>,
    "criterion_scores": {
        "criterion_1": <points awarded/deducted>,
        "criterion_2": <points awarded/deducted>,
        "criterion_3": <points awarded/deducted>,
        ...
    }
}
```

## Prompt collecting the preference-based evaluation from GPT-5 mini for PRBench

You are a financial/legal[a] expert evaluating AI responses to finance/legal questions. Your task is to compare two AI responses and determine which one is significantly better from a financial/legal perspective.

**Evaluation Criteria:**

| (if the query is related to finance) | (if the query is related to legal) |
|---|---|
| - Financial Accuracy | - Legal Accuracy |
| - Process Transparency & Auditability | - Application of Law to the Facts |
| - Handling Uncertainty | - Procedural Correctness |
| - Practical Utility | - Handling Uncertainty |
| - Risk & Ethical Disclosure | - Practical Utility |
| - Supplemental Insight | - Risk & Ethical Disclosure |
| - Instruction Following | - Supplemental Insight |
| | - Instruction Following |

**Instructions:**

1. Read the client's question carefully

2. Evaluate both Response A and Response B

3. Return ONLY a single number:

   - 1 if Response A is significantly better
   - -1 if Response B is significantly better
   - 0 if both responses are roughly equivalent in quality

**Patient Question:**
{prompt}

**Response A:**
{response_a}

**Response B:**
{response_b}

**Your evaluation (return only 1, -1, or 0):**

---

[a]Using the words "financial" or "legal" depends on the type of the corresponding query. For the A/B format below, the same logic applies.

## A.8 THE USE OF LARGE LANGUAGE MODELS (LLM)

For this project, LLMs were used to polish the writing of the main paper and to assist with coding for the numerical experiments.

