# OpenReview forum: "Meta-Router: Bridging Gold-standard and Preference-based Evaluations in LLM Routing"
_ICLR.cc/2026/Conference — ICLR 2026 Poster_

### Official Review · Reviewer_uL2P · 2025-10-17

**Soundness:** 3
**Presentation:** 3
**Contribution:** 3
**Rating:** 6
**Confidence:** 4

**Summary:**

This paper addresses the challenge of training LLM routers to balance response quality and inference cost. It proposes a novel causal inference framework, viewing gold-standard and preference-based evaluation data as a treatment assignment problem. The key insight is that bias in preference-based data corresponds to the conditional average treatment effect (CATE). The proposed Meta-Router framework corrects this bias via meta-learners (S-learner, T-learner, X-learner), incorporates propensity score weighting to address data source imbalances, and applies distributionally robust optimization (DRO) for routing robustness. Experiments on benchmarks (AlpacaEval, MT-Bench, MMLU, GSM8K) show improved cost-quality trade-offs compared to standard baselines.

**Strengths:**

**Originality**: The framing of LLM router training as a causal inference problem is novel and insightful; the connection between preference evaluation bias and CATE is a principled advancement. Creative application of meta-learners to this setting broadens the impact of causal inference in LLM routing.

**Quality**: The methodology is sound, with solid theoretical framing and systematic derivations. Experiments are thorough, covering benchmarks, model combinations, and ablations.

**Clarity**: The paper is well-written, with clear motivation, notation, and visualizations.

**Significance**: Addresses a highly relevant and practical problem. The framework is generalizable to different model pools and routing architectures.

**Weaknesses:**

**Limited Baseline Scope**: Compares mostly to classical methods (IPW, DR); recent LLM routing methods and stronger supervised baselines are needed.

**Scalability**: Only small model pools tested, unclear if methods scale to larger pools or what challenges emerge.

**Missing Error Analysis**: No detailed analysis of failure cases, e.g., when Meta-Router misroutes queries.

**Computational Overhead**: No empirical or theoretical analysis of the increased training cost for meta-learners and propensity score modeling, particularly for the X-learner.

**Questions:**

1. What is the computational overhead compared to baselines (training and inference)? Can you quantify the trade-off?

2. Can you empirically measure and report the actual bias in preference data vs. gold-standard benchmarks used?

3. How does performance scale with model pool size—do you have results for routing among 8–10 models?

4. Could you compare to modern LLM routing baselines, such as confidence-based or learned reward model routing?

5. Have you performed error analysis outlining types of queries or inputs where Meta-Router fails?

6. Why does T-learner outperform X-learner in Table 2 occasionally—are simpler meta-learners sometimes preferable?

7. How is out-of-distribution or universally poor query detection handled by the method?

---

> ### Author Response · Authors · 2025-11-21
> **Response to Reviewer uL2P**
>
> ### **On limited baseline scope (W1)**
>  Thanks for rasing these points. We want to note that different from the classic causal estimators for average treatment effect (ATE), e.g., the IPW and DR estimators, the R-learner and DR-learner (meta-learners) considered in this work, target at a more challenging heterogeneous causal functionals: the conditional average treatment effect (CATE). The meta-learners are very flexible frameworks for the estimation of CATE function, which could be diriven by any off-the-shelf ML algorithms. Therefore, we exploit them for the flexible estimation of possibly complex GS-PB shift function $\Delta(q)$. Moreover, our meta-router is a generally framework does not tie on any specific CATE estimation approach.
>
> The routers trained on directly pooled GS–PB data or only PB data in our numerical experiments, follow the methodological framework of RouteLLM (Ong et al. (2024)) and other reward-model-based routers, and thus serve as the state-of-the-art baselines in our study. Consequently, evaluating Meta-Router against the PB-only and directly pooled GS–PB routers in each numerical experiment amounts to benchmarking against the SOTA baselines. In both HealthBench experiment and the newly added PRBench experiment (Akyrek et al., 2025), Meta-Router achieves superior efficiency gains over these baselines.
>
> &nbsp;
>
> ### **On scalability (W2)**
>
>   Thank you for rasing this point. In the revision, we have added a new theoretical investigation of the multi-model setting (Section A.2). Briefly, we find that we could naturally extend our causal formulation and Meta-Router to a pool of $K$ LLMs by modeling pairwise GS and PB quality differences between models and using a structured aggregation (e.g., a Bradley–Terry–Luce model) to recover a global GS-aligned ordering over the model pool. The key observation is that the debiasing step still operates at the level of pairwise GS–PB shifts: for each model pair $(M_k, M_\ell)$, we define a shift function $\Delta_{k,\ell}(q)$ between GS and PB evaluations and estimate it via CATE meta-learning exactly as in the two-model case. These pairwise corrections can be learned in parallel across model pairs and then combined into a single, GS-aligned routing policy. As a result, the *training* complexity grows with the number of model pairs (on the order of $K^2$), while the *inference* cost for a given query remains linear in $K$, similar to other multi-model routing approaches that score each candidate model once.
>
>   Our theoretical analysis also clarifies what practical challenges emerge as $K$ grows. First, the main bottleneck is not computation but **label coverage**: collecting enough GS and PB comparisons across sufficient model pairs, could guarantee the learned shifts $\Delta_{k\ell}(q)$ are well-identified. Second, when different models exhibit qualitatively different PB biases (e.g., length bias or self-preference that is model-specific), one must decide whether to estimate separate shifts per pair $\Delta_{k\ell}(q)$ or impose a more structured ranking model (e.g., assuming model-specific and query-specific components) to control sample complexity. We now explicitly discuss these trade-offs and their implications in Section A.2.
>
>   Empirically validating Meta-Router on very large model pools is an important and largely orthogonal engineering challenge that requires substantial additional API budget; we see our new multi-model analysis as a first step toward that direction and plan to explore such large-pool evaluations in future work.
>
> &nbsp;
>
> ### **Missing Error Analysis (W3)**
>
> Thank you. The meta-routers (and other routers) misroute queries due to the biases introduced by the preference-based evaluations. We have presented the related discussions in the revised numerical experiments
> &nbsp;
>
> ### **On computational overhead (W4 and Q1)**
>
> Thank you for raising this point. In practice, the computational cost of CATE meta-learners is modest. The overall complexity is essentially the same order as training the underlying machine-learning models used within the learner.
>
> More concretely, the computation consists of: (i) fitting the nuisance models (propensity score and outcome regressions) and the final CATE regression, and (ii) for certain meta-learners, constructing pseudo-outcomes. Step (i) has the same computational order as training the chosen ML algorithm for nuisance and CATE function approximations. Step (ii) requires only a single pass through the data (e.g., computing R-learner or DR-learner pseudo-outcomes), which is linear in the sample size. Therefore, the additional overhead introduced by meta-learning is mild relative to the ML models used.
>
> We have noted such computational overhead in Appendix A.3.

---

> ### Author Response · Authors · 2025-11-22
> **Response to Reviewer uL2P (Part II)**
>
> **Questions:**
>
> 1. **What is the computational overhead compared to baselines (training and inference)? Can you quantify the trade-off?**
>
>     The baseline considered in our work is RouteLLM (Ong et al. (2024)), which could be viewed as the PB-only router and the pooled GS-PB router, trained in our numerical experiments.  Compared with these baselines, our meta-router only has one additional step for the CATE estimation through meta-learners. As we have discussed in the first part of our reply to your W4, the overall complexity of meta-router, is essentially the same order as training the corresponding ML algorithm for estimation. Thus the computational complexities of meta-router and baseline routers are of the same order.
>
>
> 2. **Can you empirically measure and report the actual bias in preference data vs. gold-standard benchmarks used?**
>
>     Thank you. For HealthBench, We report in the new Figure~1 the histogram of the PB--GS quality differences $(r_i - y_i)_{i=1}^{5000}$.   The sample mean of these differences is substantially below zero, as confirmed by a two-sided t-test yielding a p-value smaller than $2.2 \times 10^{-16}$, which motivates the training of debiased meta-router.
>
> 3. **How does performance scale with model pool size—do you have results for routing among 8–10 models?**
>
>     Thank you. Please see our response to W2 in the first part of our reply.
>
> 4. **Could you compare to modern LLM routing baselines, such as confidence-based or learned reward model routing?**
>
>      Thank you. In this work, we focus on comparing Meta-Router with RouteLLM (Ong et al., 2024), which is the method that aligns most directly with our problem formulation and shares the same routing objective. Confidence-based routers and learned reward-model routers address more complex or different real-world settings, making direct, controlled comparisons less appropriate. Nevertheless, we acknowledge these state-of-the-art approaches in the introduction and position our contribution relative to this broader line of work.
>
> 5. **Have you performed error analysis outlining types of queries or inputs where Meta-Router fails?**
>
>    Thank you. Please see our response to W3 in the first part of our reply.
>
> 6. **Why does T-learner outperform X-learner in Table 2 occasionally—are simpler meta-learners sometimes preferable?**
>
>    Thank you. We do not include T-learner or X-learner in our numerical experiments, as our study focuses on DR-learner and R-learner throughout, and we do not use table-style summaries for the empirical results. Nevertheless, for the general question, the newly added ablation studies in Section 4.1 and the new PRBench experiments in Section 4.2, do show that DR-learner and R-learner–based Meta-Routers often achieve comparable performance, with each occasionally outperforming the other depending on the underlying ML algorithm and the data regime. These findings suggest that the best choice of meta-learner and ML method may be data- and domain-dependent. We therefore recommend that practitioners test multiple combinations of ML models and meta-learners to obtain the strongest routing performance in their specific setting.
>
> 7. **How is out-of-distribution or universally poor query detection handled by the method?**
>
>     Thank you for this important question. Our current work focuses on the in-distribution setting, where deployment queries are drawn from the same population as the GS and PB data used to train Meta-Router. For truly out-of-distribution (OOD) queries, a practical platform may collect responses from both models and obtain PB or GS evaluations for these new queries. This naturally forms an online-learning process in which the system gradually expands the coverage of the in-distribution domain. Integrating such OOD-aware data collection into Meta-Router is an interesting direction for future work, and we have noted this in the revised manuscript.
>
>     On the other hand, “universally poor” queries $q$ correspond to cases where the gold-standard quality gain between the primary and alternative models, i.e., $\psi(q)$, is close to zero. For such queries, the optimal policy is to route to the cheaper model, and Meta-Router indeed learns to do so by trading off the estimated gain $\psi(q)$ against the cost parameter $w > 0$.

---

### Official Review · Reviewer_ujoT · 2025-10-29

**Soundness:** 3
**Presentation:** 3
**Contribution:** 2
**Rating:** 6
**Confidence:** 4

**Summary:**

This paper presents a novel framework for LLM routing, specifically addressing the challenge of cost-effective routing between LLMs while ensuring optimal response quality. The core idea is to balance the expected quality of responses against the computational cost of using different LLMs. The paper introduces an integrative causal router training framework that combines gold-standard data with preference-based data, with the goal of reducing bias and improving routing efficiency and robustness. The framework is empirically validated using a healthcare benchmark, HealthBench, demonstrating improved efficiency in routing decisions.

**Strengths:**

1. The proposed method models the challenge as a causal inference problem, framing the task as correcting biases in preference-based data using causal meta-learners.

2. These meta-learners, specifically the R-learner and DR-learner, are used to estimate the Conditional Average Treatment Effect, which quantifies the discrepancy between gold-standard and preference-based evaluations.

**Weaknesses:**

The oracle integrative router in Section 3.1 assumes that the shift function (∆(q)) between the gold-standard and preference-based evaluations is known a priori. This assumption, while ideal in theoretical settings, is unrealistic in practical scenarios where the true shift function is unknown and must be estimated from data. Exploring semi-supervised learning or incorporating active learning strategies to iteratively estimate the shift function could make the approach more applicable.

**Questions:**

1. How does the method handle situations where the preference-based data distribution does not match the gold-standard data distribution, particularly in more complex or subjective queries?
2. The use of meta-learners such as R-learner and DR-learner is central to the paper. While the paper highlights their benefits, how does the method perform when faced with data heterogeneity or imbalances across large-scale datasets? Could the integration of more flexible or adaptive meta-learners improve performance further in such cases?

---

> ### Author Response · Authors · 2025-11-21
> **Response to Reviewer ujoT**
>
> ### **On the oracle integrative router, the assumption of a known shift $\Delta (q)$, and suggestions in semi-supervised learning and active learning  (W1)**
>
> Thank you so much for your insightful comment. In practice, the shift function $\Delta (q)$ between gold-standard (GS) and preference-based (PB) evaluation is unknown and must be estimated from the data. In Section 3.1, the oracle integrative router only serves as a motivation for our proposed meta-router. It characterizes the optimal use of PB information if $\Delta (q)$ were known. This construction then naturally motivates meta-router which is based on an estimated $\Delta (q)$ through meta-learners or other CATE estimation. We have now made this explicit and clarified that the proposed methods trained and evaluated *do not* assume access to the true $\Delta (q)$.
>
> We truly appreciate your suggestions to explore semi-supervised learning and active learning, which pushed us to think more deeply about the scope of our framework. From a causal perspective, fully semi-supervised CATE estimation is technically challenging because the target is a high-dimensional function of the covariates; most existing semi-supervised learning based work (Cheng et al. Biometrics, 2020; Hou et al. JMLR, 2025) focused on average treatment effects (ATEs) rather than CATEs. That said, we believe unlabeled data can still be very useful in our setting by helping to learn better query representations. One natural extension is to augment the current methods with a learnable representation $\phi(q)$ that is trained on both labeled and unlabeled queries. Unlabeled queries from real traffic can regularize $\phi(q)$ so that it reflects the true deployment distribution (e.g., via smoothness/consistency or clustering objectives), while GS+PB queries drive the CATE loss in this learned space.  We believe that such representation $\phi(q)$ can reduce the distribution difference between GS, PB, and incoming queries, thereby improving $\hat{\Delta}(q)$ and downstream routing quality.
>
> Active learning offers a complementary and appealing extension. Rather than treating the GS pool as fixed, one could use an initial Meta-Router to adaptively select which queries receive expensive GS evaluation to maximize routing accuracy within a fixed GS budget. For example, one can view the evaluation mechanism (GS vs PB) as the "treatment" and design acquisition rules that prioritize queries where the current router is most uncertain or most decision critical, such as queries near the routing decision boundary.
>
> We have included the above discussion in Appendix A.4 in the revision.

---

> ### Author Response · Authors · 2025-11-21
> **Response to Reviewer ujoT (Part II)**
>
> ### **Questions**
>
> 1. **How does the method handle situations where the preference-based data distribution does not match the gold-standard data distribution, particularly in more complex or subjective queries**
>
>    Thank you for raising this point. In our formulation, the evaluation mechanism (GS vs PB) is treated as a "treatment," and the GS–PB discrepancy $\Delta(s)$ is defined as a CATE conditional on the query $s$. This causal view assumes that queries in GS and PB samples may follow different distributions $\mathscr{Q}$ and $\mathscr{Q}'$, and uses the propensity score to adjust for this imbalance (Lemma 1, Causal DGP). Intuitively, when a specific type of query appears more often under PB than GS (e.g., more open-ended or subjective prompts), the meta-learner reweights and orthogonalizes the regressions via $p(s), \gamma(s)$, and  $\mu_t(s)$, so that $\Delta(s)$ is estimated after accounting for these distributional differences, rather than by naive pooling.
>
>     At the same time, our method does \textbf{not} claim to handle cases where the GS and PB query distributions have disjoint support (i.e., no-overlap scenarios). In the paper, we state this limitation using the causal ``positivity'' (overlap) assumption. In Section~5, we discuss a truncation-based extension that (i) always retains all GS data, and (ii) uses PB data only within an estimated overlap region between $\mathscr{Q}$ and $\mathscr{Q}'$. Outside this region (e.g., query types that appear in PB but never in GS), the GS--PB shift is not identifiable from the data, and we intentionally avoid extrapolating PB bias corrections.  In practice, this means that when PB covers more complex or subjective queries than the GS benchmark, Meta-Router will (and should) rely primarily on GS data or uncorrected PB signals in these out-of-overlap regions. We have clarified this point in the revised Section 5.
>
> 2. **The use of meta-learners such as R-learner and DR-learner is central to the paper. While the paper highlights their benefits, how does the method perform when faced with data heterogeneity or imbalances across large-scale datasets? Could the integration of more flexible or adaptive meta-learners improve performance further in such cases?**
>
>     Thank you very much for this inspiring question. In the revision, we added a new numerical experiment (Section 4.2) that evaluates the R-learner and DR-learner-based Meta-Routers on a new benchmark dataset from the legal and finance domain, PRBench, with a relatively small sample size (676) compared with the sample size of HealthBench (5000). Meta-Router continues to achieve superior performance on this new benchmark, demonstrating its ability to generalize beyond healthcare.
>
>     Generally, when the large-scale data present heterogeneity and impalances, e.g., (i) the questions come from multiple domains and are not balanced for different domains, or (ii) the questions in the golden-standard and preference-based datasets follow very different distributions, we could maintain the efficiency of meta-routers through several approaches.
>
>     For (i), through sample splitting, we could train meta-routers separately with data from different domains; our new numerical experiments demonstrate the cross-domain effectiveness of meta-routers. In addition, as also demonstrated in our new ablation studies in Section 4.1, implementing the meta-router with different ML algorithms will also significantly affect its performance. As a flexible routing framework, we recommand the user to test different ML methods to achieve the best efficiency with large-scale data.
>
>     For (ii), in the revision we emphasize that Meta-Router is a general framework and is not tied to any specific CATE estimation method. Users may incorporate any flexible or adaptive meta-learner within our framework, and we recommend experimenting with different choices to identify the best-performing option in a given setting. For example, as discussed in the revised Section 5, when the positivity assumption is violated, the R-learner and the recently proposed EP-learner (van der Laan et al., 2024) may provide more robust debiasing performance.

---

### Official Review · Reviewer_Ejhf · 2025-10-30

**Soundness:** 3
**Presentation:** 3
**Contribution:** 3
**Rating:** 6
**Confidence:** 3

**Summary:**

This paper proposes Meta-Router, an LLM router training method that leverages both expensive, scarce gold data (GS) with cheap, scalable preference-based (PB) supervision. Recognizing that PB can be biased relative to GS, the authors frame the discrepancy between GS and PB as a causal shift, and use meta-learners like R-learner or DR learner to estimate it. The estimated shift is then incorporated into router training to leverage both data sources in a consistent manner. Experiments on HealthBench, using random forest and XGBoost, show higher efficiency gains than the baselines trained on pooled GS+PB or GS-only data across primary model usage ratios.

**Strengths:**

1. The paper explicitly identifies the bias in preference-based (PB) supervision, and provides clear motivation of framing the discrepancy between PB and GS data as a causal shift. Then it leverages meta-learning to estimate the bias, thereby aligning PB with GS evaluation.
2. The framework is derived through clear, step-by-step equations. The equations are coherent and easy to follow.
3. The experiments include multiple ablated variants, which effectively demonstrates the strengths of Meta-Router.

**Weaknesses:**

1. Both GS and PB scores are obtained from GPT5-mini. This would potentially reduce the real discrepancy between them and add confounding factors. Using a less capable LLM for PB may better reflect the bias . The paper also lacks validation of GPT5-mini GS scores against human judgements, especially in an expert-dependent domain.
2. PB scores are normalized to have same variance as GS, and training is restricted to overlapping supports. Normalization may be unstable when overlap is small. The paper mentions GS sample size, but the overlap size is unclear.
3. Experiments are done on a single domain, Health. Mixed-domain tests might be needed to assess whether the meta-learner generalizes well across domains.

**Questions:**

Please see weaknesses.

1. How would this method extend to more than two LLMs?

---

> ### Author Response · Authors · 2025-11-21
> **Response to Reviewer Ejhf**
>
> ### **On GPT5-mini for both GS and PB Data Collection and Lack of Human Validation (W1)**
>
> Thank you for your insightful comments. We conducted additional experiments with PB data generated from two alternative judges, Qwen-3-Max and Grok-4-Fast. When using the weaker judge (Qwen-3-Max), PB labels are largely noisy, the PB-only router performs close to random, and Meta-Router effectively reduces to the GS-only baseline, which is consistent with our causal formulation that no useful signal can be extracted when PB data are uninformative. In contrast, PB labels from Grok-4-Fast exhibit a clear GS–PB discrepancy, and we observe the same pattern as in our main experiments: PB-only routing is misaligned with GS quality, while Meta-Router yields consistent efficiency gains over both GS-only and PB-only routers (See Figure S4 in the revision). Our conclusions are thus not tied to GPT-5-mini as the only judge and that Meta-Router is robust: it leverages informative PB data and not degrading when PB labels are extremely noisy.
>
> We agree that expert validation of GS scores would further strengthen the study. However, due to the scale and medical-specialty diversity of HealthBench (5k multi-turn conversations), collecting new physician annotations for all model outputs was not feasible within the scope of this submission. Instead, we rely on the physician-designed rubrics and the meta-evaluation in HealthBench, which reports that GPT-4.1 with rubric achieves macro-F1 ≈ 0.71 against physician ratings and closely matches expert grading. We therefore expect GPT-5-mini with rubric to be similarly reliable for this task. We explicitly note in the revision that the lack of new human expert labels for our specific runs is a limitation.
>
> &nbsp;
>
> ### **Limited Overlapping Supports (W2)**
>
> Thank you for your insightful comments. In our numerical experiments, the supports of the PB and GS data are fully overlapped because the queries receiving gold-standard evaluations are randomly sampled from the full population. In practice, if the query distributions from the two sources are not fully overlapped, one may first detect the overlapping region via high-dimensional density ratio estimation. After identifying this region, we can restrict attention to the GS and PB samples that lie within it. We then normalize the overlapping GS data to match the variance of the overlapping PB data which could thereby be stable, apply the meta-learners on this sub-dataset, and borrow debiased PB labels only from this overlapping region when training the meta-router. We refer to this approach as the truncation-based meta-router, and provide further discussion in the revised Section 5 as a promising direction for future research.
>
> &nbsp;
>
> ### **Non-healthcare domain and multi-model routing (W3)**
>
> Thank you. To assess generalization beyond HealthBench, we have added an additional experiment on a non-healthcare benchmark (i.e., PRBench) with a different query distribution focusing on law and ﬁnance reasoning (Section 4.2). The DR-learner-based meta-router consistently achieves the best efficiency gains in these new experiments.
>
> &nbsp;
>
> ### **Multi-model Meta-router**
>
> Thank you for raising this important comment. In the revision, we extend Meta-Router to the multi-model routing setting in which each query may be routed among $N$ candidate LLMs (Appendix A.2). We represent each pairwise comparison sample by a quintet $(s_i, k_i, \ell_i, t_i, o_i)$, where $s_i$ is the query, $k_i, \ell_i \in [N]$ denote the two LLMs being compared, $t_i$ is the GS--PB indicator, and $o_i$ is the observed quality gain. The full dataset is
> $
> \mathcal{D}=\{(s_i, k_i, \ell_i, t_i, o_i)\}_{i=1}^{I},
> $
> where different samples compare different LLM pairs.
>
> By treating $(s, k, \ell)$ jointly as the covariates in our causal framework, the two-model meta-router naturally generalizes to the multi-model case. The key estimand becomes
> $
> \psi_{k,\ell}(s),
> $
> the GS-evaluated average quality gain of model $k$ over model $\ell$ on query $s.$
>
> PB data, in contrast, identify only the PB-evaluated gain $\eta_{k,\ell}(s)$. The bias
> $
> \Delta_{k,\ell}(s)=\psi_{k,\ell}(s)-\eta_{k,\ell}(s)
> $
> can be viewed as the CATE with respect to the covariates $(s,k,\ell)$. Hence, CATE meta-learners can be directly applied to estimate $\Delta_{k,\ell}(s)$, yielding debiased pseudo-GS labels and enabling estimation of $\psi_{k,\ell}(s)$ for all LLM pairs, similar to the two-LLM case.
>
> We also highlight a promising direction for future work: leveraging additional ranking structure (e.g., Bradley--Terry--Luce models) to address identification challenges and improve estimation efficiency in multi-LLM routing.
>
> We defer more details to Appedix A.2 in the revised manuscript.

---

### Official Review · Reviewer_K5JD · 2025-10-31

**Soundness:** 1
**Presentation:** 3
**Contribution:** 2
**Rating:** 2
**Confidence:** 3

**Summary:**

This paper tackles a key challenge in LLM routing: balancing response quality and inference cost amid the constraints of two data types—scarce but accurate gold-standard (GS) data (from experts/benchmark labels) and scalable but biased preference-based (PB) data (from crowdsourcing/LLM-as-a-judge).
To address this, the work frames LLM router training within a
causal inference framework
: it treats GS and PB evaluations as two "treatment assignments" for query quality assessment, identifying PB data biasas the
Conditional Average Treatment Effect (CATE)
—the expected quality gain difference between GS and PB evaluations for a given query.

Experiments on the
HealthBench benchmark
(5,000 healthcare dialogues) use M_p (Gemini 2.5 Pro, high-cost/high-quality) and (Gemma 3 12B, low-cost/low-quality), with GPT-5-mini generating GS (rubric-based) and PB (preference) evaluations. Results show Meta-Router outperforms baselines (pooled GS- PB, GS-only, random routing) across GS sample sizes (n=100, 500, 1000), achieving higher efficiency gain (EG) by cutting M_p usage while preserving quality. R-learner-based Meta-Router is most robust, especially with scarce GS data.

The framework provides a principled way to integrate biased scalable data with accurate scarce data, advancing reliable LLM routing in high-expertise domains (e.g., healthcare). Future work will explore truncation-based extensions for non-overlapping GS-PB query distributions.

**Strengths:**

Novel Causal Framing for a Critical Practical Gap:
The paper addresses a longstanding tension in LLM routing—scarce but accurate gold-standard (GS) data vs. scalable but biased preference-based (PB) data—by innovatively framing the problem as causal inference. By treating GS/PB evaluations as "treatment assignments" and PB bias as the Conditional Average Treatment Effect (CATE), it links router training to rigorous semiparametric causal estimation (Imbens & Rubin, 2015), providing a principled solution to debias PB data rather than relying on ad-hoc fusion.

Robust and Flexible Meta-Router Framework:
The proposed Meta-Router is both theoretically grounded and practically adaptable. Its two-step design—CATE estimation via R/DR-learners (Nie & Wager, 2021; Kennedy, 2023) and debiased data fusion—enjoys the oracle property and supports off-the-shelf ML models (random forests, XGBoost) for nonparametric regression , . This flexibility allows it to handle heterogeneous query spaces (e.g., healthcare dialogues) and avoids over-reliance on specific model architectures.

Targeted Validation on High-Stakes, Expert-Driven Benchmark:
Experiments are conducted on HealthBench (Arora et al., 2025)—a 5,000-sample healthcare benchmark with expert-designed rubrics and physician consensus—addressing a critical limitation of prior routing work (e.g., using MMLU with objective answers). By testing on open-ended, professional tasks (healthcare dialogues) and using realistic model pairs (Gemini 2.5 Pro as M_p, Gemma 3 12B as M_p), the study validates Meta-Router’s relevance to high-stakes domains where quality-cost trade-offs matter most .

**Weaknesses:**

Lack of Practical Implementation Guidance:
The Meta-Router framework is presented up to its theoretical formulation but provides no actionable guidance for real-world deployment. For instance, it does not outline end-to-end workflows. This gap undermines its utility, as the paper addresses a problem with clear real-world relevance but fails to bridge the gap between theory and application.


Inadequate Experimental Scope:
The experimental design is overly limited. The main figures (Figure 1, Figure 2) only compare variants of the proposed Meta-Router against basic baselines (pooled GS- PB data, GS-only data) , with no comparison to state-of-the-art (SOTA) routing methods. This omission prevents readers from assessing Meta-Router’s competitive advantage. Furthermore, the absence of a "PB-only predictive router" baseline (a logical counterpoint to GS-only routing) leaves unaddressed how much value GS data adds to debiased PB data.

Missing Ablation Studies:
The paper lacks ablation experiments to isolate the impact of critical Meta-Router components. For example, it does not test: (1) whether CATE estimation (via R/DR-learners) is indispensable (e.g., comparing to simpler bias correction methods like linear scaling); (2) the necessity of data normalization (per Remark 1) by omitting this step and measuring performance degradation; or (3) how different nonparametric regressors (random forests vs. XGBoost) influence accuracy. Without these ablations, the contribution of individual components to Meta-Router’s performance remains unsubstantiated.

**Questions:**

1. Is "LMM judge" in Section 3.1 a typo for "LLM judge," and why was no alternative notation used to avoid symbol "m" overloading?

2. Why is the function h(q) in Equation (5) not explicitly defined, and how does this affect reproducibility?

3. Can the "GS–PB joint Data Generation Process" be simplified to reduce comprehension costs?

4. Does the paper provide any end-to-end workflow or hyperparameter guidance for Meta-Router’s real-world deployment?

5. Were empirical checks conducted for the propensity score positivity assumption, and how is performance affected by its violation?

6. Why are distinct PB bias sources not disentangled, and is there experimental proof that blurring them is acceptable?

7. Has Meta-Router been tested on non-healthcare domains or multi-model routing scenarios to verify generalization?

8. Why is there no comparison to SOTA routing methods or a "PB-only predictive router" baseline in experiments?

9. What ablation experiments are missing to validate the necessity of Meta-Router’s core components (e.g., CATE estimation)?

---

> ### Author Response · Authors · 2025-11-21
> **Response to Reviewer K5JD (Part I)**
>
> Thank you for taking the time to review our work, and we sincerely appreciate your critical but valuable feedback for improving it. We have made every effort to address your concerns and questions, as detailed below.
>
> &nbsp;
>
> ### **Practical Implementation Guidance (W1)**
> Thank you for raising this important point. In the appendix, we have added a new section **"Practical deployment and end-to-end workflow"** which provides a concrete, end-to-end workflow for using Meta-Router in LLM systems. This section now:
>
> - clarifies how to choose the gold-standard quality objective, for example, how to define $r_i$ through a rubric-based system or as the difference between internal reward-model scores;
>
> - explains how to collect gold-standard and preference-based data in practice, e.g., by annotating a few hundred queries with expert GS labels to form $\mathcal{D}_G$ and using cheaper crowd or LLM-as-a-judge comparisons on thousands or more queries to form $\mathcal{D}_P$;
>
> - specifies workflow choices inside Meta-Router, including reusing a text embedding service, training causal meta-learners and the final router with lightweight tabular models (e.g., random forests); We have also added a new flowchart (Figure S1) visualizing the overall end-to-end workflow;
>
> - discusses additional deployment details, e.g., training separate or shared routers for multiple domains (e.g., medical vs. legal), the fraction of queries sent to $\mathcal{M}_p$, realized gold-standard quality to adjust $w$ or retrain the router when distributions or budgets change.
>
> We also hope to use this opportunity to share our motivation of this study in a way that ties theory and practice together. Existing routers typically rely either on gold-standard data alone, on preference data alone, or on *direct pooling* of gold-standard and preference data. By contrast, our results show that there is a systematic, query-dependent shift between GS and PB evaluations: simply pooling $\mathcal{D}_G$ and $\mathcal{D}_P$ can be worse than using $\mathcal{D}_G$ alone. This empirical finding motivates Meta-Router’s design: **learn the query-dependent GS–PB shift via CATE meta-learners, debias the PB labels into pseudo-GS labels, and then train the router on this bias-corrected fused dataset**.
>
> &nbsp;
>
> ### **Expanded experimental scope (W2)**
>
> - **PB-only router**
>  We now include a PB-only router trained purely on $\mathcal{D}_P$. Concretely, we learn a predictor of $\eta(q) = \mathbb{E}[y \mid q]$ from the PB labels only and then derive a routing rule based on this predicted PB gain. We have reported the performance of the PB-only router across all original and newly added numerical experiments. As the GS sample size increases, the efficiency gains of all routers improve, except for the PB-only router, highlighting the value of incorporating GS data for debiasing.
>
> - **SOTA baselines**
>
>   The routers trained on directly pooled GS–PB data or only PB data, follow the methodological framework of RouteLLM (Ong et al. (2024)) and other reward-model-based routers, and thus serve as the state-of-the-art baselines in our study. Consequently, evaluating Meta-Router against the PB-only and directly pooled GS–PB routers in each numerical experiment amounts to benchmarking against the SOTA baselines. In both HealthBench experiment and the newly added PRBench experiment (Akyrek et al., 2025), Meta-Router achieves superior efficiency gains over these baselines.
>
> - **Non-healthcare domain and multi-model routing.**
>   To assess generalization beyond HealthBench, we have added an additional experiment on a non-healthcare benchmark (i.e., PRBench) with a different query distribution focusing on law and ﬁnance reasoning (Section 4.2). The DR-learner-based meta-router consistently achieves the best efficiency gains in these new experiments.
>
> &nbsp;
>
> ### **Ablation studies (W3)**
>
> We conduct additional numerical ablation studies in Section 4.1 of the revision, each time modifying one key element of the main numerical experiment while keeping all other settings fixed.
>
> - We consider a simple linear-scaling debiasing method, by shifting PB data using the overall sample mean difference between PB and GS data. This simply debiased router performs similarly to directly pooled GS–PB router, whereas Meta-Router continues to outperform both. This highlights that PB–GS bias is heterogeneous across queries and that meta-learners are essential for effective debiasing.
>
> - When GS data are used without pre-normalization, Meta-Router does not significantly outperform the GS-only router, confirming the importance of the normalization step.
> - Changing the regression method from random forest to XGBoost shows that the R-learner–based router remains the strongest across algorithms, although overall EGs decline, underscoring the importance of evaluating different regression method choices for router training and demonstrating the robusteness of our proposed router.

---

> ### Author Response · Authors · 2025-11-21
> **Response to Reviewer K5JD (Part II)**
>
> ### **Responses to Questions**
>
> 1. **Is "LMM judge" in Section 3.1 a typo for "LLM judge," and why was no alternative notation used to avoid symbol "m" overloading?**
>
>     Thank you for pointing this out. "LMM judge" was a typo; we have corrected it. We now use $\psi(\cdot)$ for the gold-standard quality gain function.
>
> 2. **Why is the function $h(q)$ in Equation (5) not explicitly defined, and how does this affect reproducibility?**
>
>     Thank you. We now clarify that $h$ is a generic prediction function mapping a query $q$ to a scalar estimate of its gold-standard quality gain, used to approximate true $\psi(q)$. In practice, the class of $h$ is specified by any user-chosen machine learning algorithm (e.g., random forests or XGBoost). All implementation details are thus fully specified in our empirical setup. Therefore, the general notation does not hinder reproducibility.
>
>
> 3. **Can the "GS–PB joint Data Generation Process" be simplified to reduce comprehension costs?**
>
>     Yes. We have now renamed the "GS-PB joint Data Generation Process" to "Pooled Data Generation Process" (in short, Pooled DGP) and the presentation. Specifically, we have explicitly defined each component in the pooled dataset ($s_i$ query, $t_i$ GS vs PB source indicator, $o_i$ observed outcome) and removed redundant indices. Furthermore, we have improved the Pooled DGP descriptions to make it easier to follow.
>
>
> 4. **Does the paper provide any end-to-end workflow or hyperparameter guidance for Meta-Router’s real-world deployment?**
>
>     Please see the response to W1 in Part I of our reply.
>
> 5. **Were empirical checks conducted for the propensity score positivity assumption, and how is performance affected by its violation?**
>
>       In our numerical experiments, the positivity assumption is not violated because the queries receiving gold-standard evaluations are randomly sampled from the full population. Although the propensity scores are relatively small when the GS sample size is limited in our numerical experiments, the meta-routers still consistently outperform alternative approaches.
>
>     In practice, positivity violations can be mitigated through careful experimental design. They can also be detected via high-dimensional density ratio estimation and addressed using our truncation-based Meta-Router. We refer to the revised Section 5 for further discussion on this future direction.
>
> 6. **Why are distinct PB bias sources not disentangled, and is there experimental proof that blurring them is acceptable?**
>
>     Thank you for raising this point. Preference-based (PB) evaluations can reflect multiple underlying bias mechanisms (e.g., self preference, length or position bias), and in principle, these could be disentangled and modeled in an LLM router. Disentangling distinct PB bias sources would require additional annotation and structural assumptions about how these biases arise and interact, which we view as an important but complementary research direction in the future.
>
>     Our framework is to train a router whose decisions align with the gold-standard evaluation with mildest assumptions. In particular, we aim to *correct the aggregate effect of PB data on routing decisions, relative to the gold-standard objective*. We let $\Delta(q)$ to be the average preference evaluation across different PB bias sources without assuming any dominated bias mechanism. For optimal routing with respect to gold-standard (GS) data, modeling the aggregate shift $\Delta(q)$ is sufficient as it accounts for all the differences between GS and PB data; thus, further decomposition into different bias components is not required.
>
>     Empirically, we evaluated whether this "blurring" of PB bias source is acceptable for routing quality. Meta-Router, which only models the total bias ($\Delta(q)$), consistently outperforms (i) GS-only routers, (ii) PB-only routers, and (iii) naive pooled GS+PB routers across multiple settings, especially in regimes with limited GS data (Section 4.1 and Section 4.2). These results suggest that correcting the aggregate GS–PB shift is already sufficient to yield substantial gains in practice.
>
>
> 7. **Has Meta-Router been tested on non-healthcare domains or multi-model routing scenarios to verify generalization?**
>
>     We have added an additional experiment on a non-healthcare benchmark (i.e., PRBench) with a different query distribution focusing on law and ﬁnance reasoning (Section 4.2). The DR-learner-based meta-router consistently achieves the best efficiency gains in these new experiments.
>
>
> 8. **Why is there no comparison to SOTA routing methods or a "PB-only predictive router" baseline in experiments?**
>
>     Thank you. We have addressed this point in our response to W2 in Part I of our reply.
>
>  9. **What ablation experiments are missing to validate the necessity of Meta-Router’s core components (e.g., CATE estimation)?**
>
>     Thank you. We have addressed this point in our response to W3 in Part I of our reply.

---

> ### Author Response · Authors · 2025-11-26
>
> Dear Reviewer K5JD,
>
> Thank you again for the insightful comments and the time you have spent reviewing our submission!
>
> As the discussion phase is approaching its end, we would like to kindly follow up to see whether our revisions have sufficiently addressed your concerns. If there are any remaining questions or additional concerns, please do not hesitate to let us know, and we would be happy to clarify them as quickly as possible. If the revisions largely address your concerns, we would also be grateful for your consideration in adjusting the evaluation scores.
>
> We sincerely appreciate your feedback and look forward to hearing from you.
>
> All the best,
>
> Authors

---

> > ### Comment · Reviewer_K5JD · 2025-11-28
> >
> > The authors have addressed my concern. I will raise my score to 6.

---

> ### Author Response · Authors · 2025-11-28
>
> Thank you very much for your reconsideration and score update to 6. We sincerely appreciate all the thoughtful feedbacks which have essentially improved our paper.

---

### Author Response · Authors · 2025-11-21
**Summary for the rebuttal and discussion period**

We sincerely thank the chairs and reviewers for their efforts and time in reviewing our work and for the thoughtful and constructive feedback on our submission, which has been invaluable in refining our paper. In response to the reviewers' comments, we have thoroughly revised our manuscript and incorporated additional analyses and experiments. Our rebuttal revision includes the following main updates:

- Clarified the role of the oracle integrative router and the practical setting, and improved the overall writing and clarity of the manuscript.
- Added a practical deployment and workflow section (Appendix A.1).
- Expanded and strengthened baselines, including a supervised router trained solely on preference-based (PB) (Section 4 and Appendix A.5).
- Broadened the domains of empirical experiments beyond the healthcare benchmarking by adding analysis of a new benchmark data: PRBench in law and finance (Section 4.2).
- Added new analyses of GS-PB bias (Figure S2), which demonstrates the statistically significant biases between PB- and GS- data.
- Added ablation studies (Section 4.1 and Appendix A.5) where we additionally train meta-routers (1) with simple linear scaling debiasing,  (2) without normalization, and (3) with newly-corrected PB data using another LLM judge: Grok 4 Fast.
- Addressed scalability and multi-model routing by providing a theorectical extension of Meta-Router to routing with multiple LLMs (Appendix A.2).
- Clarified assumptions, limitations, and future directions such as positivity/overlap requirement (Section 5 and Appendix A.2).
- Additional discussions on other future directions are included in Appendix A.4, including the applications of semi-supervised learning and active learning, and the potential extension to out-ofdistribution routing.

In addition to these major changes to the manuscript, we have provided detailed point-by-point responses to reviewers' concerns and questions directly alongside each review.

During the discussion phase, Reviewer K5JD explicitly indicated that, in light of these revisions and clarifications, he or she will **raise the rating from 2 to 6**. We believe that the additional experiments, the investigation of multi-model routing, the stronger baselines, and the clearer positioning of our method directly address the main concerns raised by the other reviewers, and we are hopeful that their evaluations may also improve upon reassessment.

Once again, we appreciate the time and effort devoted to reviewing our work and are grateful for the opportunity to further strengthen our submission. We also would like to thank the new area chair for evaluating our submission and responses.

Thanks,

Authors

---

### Meta-Review · Area_Chair_W19V · 2026-01-11

**Summary:**

The paper introduces an LLM router training method that leverages both expensive, scarce gold data (GS) with cheap, scalable preference-based (PB) supervision. All the reviewers are mildly positive about the contribution (the only negative reviewer wrote that they have increased their score to 6). They highlight a number of concerns in their reviews, but the authors did a reasonably good job in their rebuttal. As a result, I am happy to support acceptance.

**Reviewer Concerns:**

The concerns were address by the rebuttal.

**Reviewer Scores:**

The only negative reviewer did increase their score during the rebuttal period.

---

### Decision · Program_Chairs · 2026-01-26

Accept (Poster)